# Preliminary Assessment of Ecological Status of the Siversky Donets River Basin (Ukraine) Based on Phytoplankton Parameters and Its Verification by Other Biological Data

Olena Bilous [1], Sergey Afanasyev [1], Olena Lietytska [1], Oksana Manturova [1], Oleksandr Polishchuk [2], Inna Nezbrytska [1], Maryna Pohorielova [1] and Sophia Barinova [3,*]

[1] Institute of Hydrobiology of the National Academy of Sciences of Ukraine, Geroiv Stalingrada Ave., 12, 04210 Kyiv, Ukraine; bilous_olena@ukr.net (O.B.); safanasyev@ukr.net (S.A.); lietitska@ukr.net (O.L.); omanturova@ukr.net (O.M.); inna_imn@ukr.net (I.N.); chertkovams1988@gmail.com (M.P.)
[2] M.G. Kholodny Institute of Botany of the National Academy of Sciences of Ukraine, Tereshchenkivska St., 2, 01601 Kyiv, Ukraine; mrpolishchuk@gmail.com
[3] Institute of Evolution, University of Haifa, Mount Carmel, Abba Khoushi Ave. 199, Haifa 3498838, Israel
* Correspondence: sophia@evo.haifa.ac.il; Tel.: +972-4824-97-99

**Abstract:** The river basin of Siversky Donets is of great scientific interest since this river runs through a territory with heavy industry (in particular, coal mining, chemical processing and metal industries). Within the basin, rivers of different sizes were explored (small, medium, large and extra-large) that flow through siliceous and calcareous rocks on the same elevation (lowland—below 200 m a.s.l.). Phytoplankton, as one of the Biological Quality Element, was used to perform the assessment of ecological status of the water bodies within the Siversky Donets river basin in 2019. The state monitoring program based on the updated approaches has been implemented in the river basin for the first time. The composition of phytoplankton species in the basin comprised 167 species (168 intraspecies taxa), mainly Bacillariophyta (63%) and Chlorophyta (22%) with the presence of other species (Cyanobacteria, Charophyta, Chrysophyta, Dinophyta and Euglenophyta). High species diversity and divisions amount are a distinctive property of the smaller rivers, while the bigger rivers show lower number of divisions. The "bloom" events, which are important ecological factors, were not detected in the Siversky Donets river basin. Algal species composition in plankton samples of the basin was identified and series of ecological parameters, such as habitat preferences, temperature, pH, salinity, oxygenation and organic water pollution according to Watanabe and Sládeček's index of saprobity (S) trophic state and nitrogen uptake metabolism were analyzed. The ecological conclusions were also verified by a canonical correspondence analysis (CCA). The significance of the Canonical Correspondence Analysis (CCA) results was estimated of by a Monte-Carlo permutation test. The high concentrations of inorganic phosphorus compounds (permanganate index (CODMn)) and nitrite ions favored the diversity of Chlorophyta and Cyanobacteria diversity correlated with the levels of bicarbonate and CODMn. High diversity of diatoms was facilitated by the total amount of dissolved solids and chemical oxygen demand (COD). It was found that low water quality could be associated with conditions leading to predominant growth of the mentioned groups of algae. According to the analysis, the highest water quality was characterized by balanced phytoplankton composition and optimal values of the environmental variables. The sites with reference conditions are proposed for future monitoring.

**Keywords:** water quality assessment; monitoring; river; algae; water bodies





## 1. Introduction

River systems belong to the most open dynamic ecosystems and are characterized by a fairly active interaction between the biota of the channel and the floodplain; additionally, they play an important role as a habitat for a huge number of living organisms [1,2]. Large

rivers serve as biodiversity reserves and areas of speciation processes [3]. They provide various ecosystem services for society, such as supplying humans and animals with fresh drinking water and serving as waterways and sources of hydro-energy, fisheries and recreational activities [4,5].

Notwithstanding the high importance of the rivers, most of the river basins are threateningly affected by pollution, channel modifications, inter-basin water transfers and modified flow regimes, fishing pressure, deforestation, agriculture, urbanization and climate change [6–11].

To prevent river ecosystems' degradation and maintain an appropriate ecological state, there is an urgent need to determine factors and processes affecting river basins on the whole and at regional scales.

One of the river basins with strong research interest is the Sivesky Donets, which runs mainly in the south-eastern part of Ukraine. The region is known for its heavy industry, in particular coal mining, chemical processing and metal industry. The intensive mining led to significant environmental damage. In addition, this basin has been in the zone of an armed conflict between Ukraine and Russia since 2014, which in turn determines the task of the correct assess of the ecological state and biological resources for the subsequent determination of damage. The mentioned factors provoked a drastic ecological situation and posed a range of new risks, mainly for the water ecosystems of the region. The relevance of this issue is explained by the river's role in the region as the river and its tributaries provide 80–85% of the water taken by the main water provider (the Donbas Water Company). The vast majority of this water comes from the surface runoff of the rivers in the area, tainting river beds, canals and water reservoirs. Military conflict and pollution by industrial and municipal enterprises endanger the lives of civilians which use this water [12,13].

Ukraine started to implement the European Water Framework Directive [14] in 2014 by restructuring the monitoring system. The Siversky Donets river with its tributaries was chosen as the first river basin where the state monitoring based on the updated approaches has been implemented. According to the modern approaches to water resources management, their characterization should be based on the basin principle [15], the priority role is given to the biotic component, which, along with hydromorphological and hydrochemical indicators, can be used to establish the ecological status of a water body within a certain basin. As one of the biological quality elements, phytoplankton is proposed as a sensitive indicator of the environmental changes in the aquatic ecosystems, including changes in hydrological conditions, nutrient loads and other environmental conditions [9,16–24]. Thus, the aim for this work was to analyze available information to find reliable unpolluted sites in the Siversky Donets river basin, which can be used as a reference sites. Additionally, the purpose of the work is the characteristics of phytoplankton of these sites with the subsequent assessment of the ecological state in the basin according to phytoplankton parameters and to verify this assessment by the other biological indicators.

## 2. Materials and Methods

### 2.1. Study Area

The Sivesky Donets river is one of the largest rivers in the eastern part of Ukraine and the longest tributary of the Don river. Its total length is ca. 1053 km, and the catchment area is equal to 98.9 thousand km$^2$. The basin comprises 1489 small rivers with a total length of 8.8 thousand km, of which 11 are above 100 km long. The average drainage density is 0.17 km$^{-1}$ [25].

The river arises in the Belgorod district of the Russian Federation, and on its 944th km, it enters Ukraine and returns to the FRF on the 220th km. Within the territory of Ukraine, its length is 724 km; the Ukrainian part of the basin is ca. 54.5 thousand km$^2$, which is 9.1% of the whole territory of Ukraine [25,26]. The river system comprises an extensive system of tributaries. The largest tributaries are the Oskil river (length is 472 km), Aidar

river (264 km), Lugan' river (198 km), Derkul river (163 km) and Kazennyi Torets river (134 km) [27].

The basin of the Siversky Donets river within Ukraine is characterized by a variety of natural conditions and significant anthropogenic impact [12]. The climate of the basin is arid continental with hot summers and cold winters. Relative humidity reaches its maximum in December (86%) and minimum in May (60%). The average annual humidity is 74%. Mean temperatures vary from 20 °C in July–August to 7 °C in January, respectively. The average annual precipitation is 525 mm; however, the amount is different for the different parts of the river basin [27]. The geological structure of the Siversky Donets river basin is a complex structure. The crystalline base of the platform consists of gneisses, shales and granites. Deep deposits are covered with thick layers of sand, clay, marl, sandstone, siltstone and chalk that were formed during the existence the sea basin in this area [25].

## 2.2. Sampling Strategy

Over a peak of plant vegetation, in July–August of 2019, the field trip to different survey areas in the Siversky Donets river basin took place. The sampling sites cover the basin of the Siversky Donets uniformly and were located within minimum disturbed landscape (including natural parks or nature reserves). Among the factors considered during the selection procedure were absence (which was essential to have physical impossibility of landscaping due to the terrain) or the minimal area of agricultural land and settlements and a certain distance from settlements and from unregulated dams. Altogether, 24 sites were explored: 11 in Luhansk district, 5 in the Dotensk region and 8 in the Kharkiv region (Figure 1).

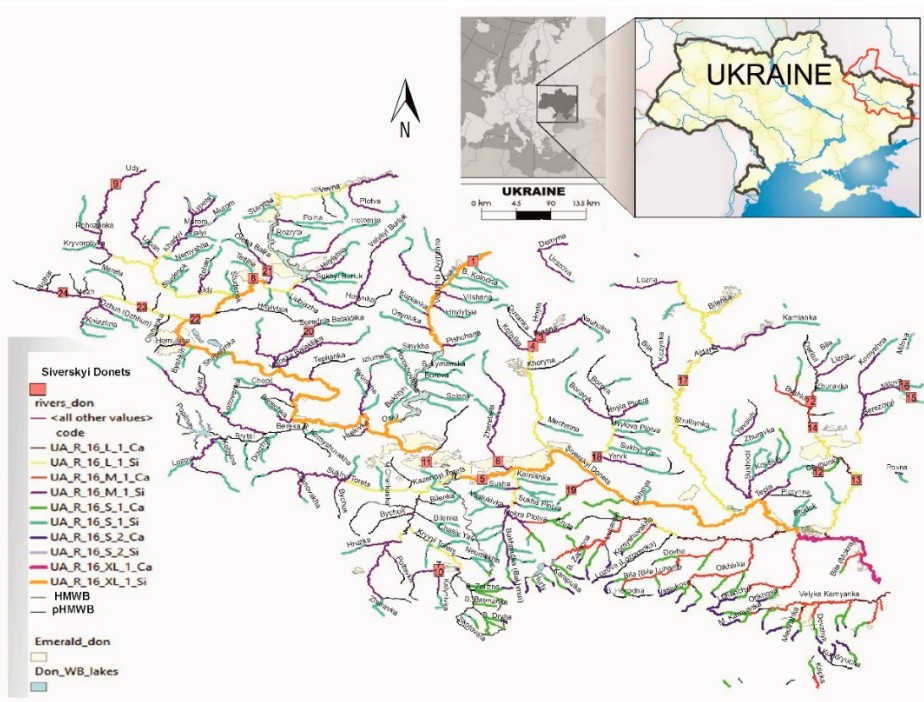

**Figure 1.** Map of the Siversky Donets river basin (Ukrainian part) with types of water bodies within the basin: 1—Oskil river, 2—Derkul river (near Bilovodsk village); 3—Krasna river (near N. Oduvanya village), 4—Krasna river (near Preobrazhennya village), 5—Siversky Donets river (near Kryva Luka village), 6—Zherebets river, 7—Bychok river, 8—Tetliha river, 9—Udy river, 10—Bychok river, 11—Siversky Donets river (near Mayaky village), 12—Chuhynka river, 13—Derkul river (near Krasnyy Derkul village), 14—Derkul river (near Novoderkul village), 15—Cherepakha river, 16—Milova river, 17—Aidar river, 18—Borova river, 19—Bilenka river, 20—Serednia Balakliika river, 21—Velyka Babka river, 22—Siversky Donets river (near Cheremushne village), 23—Mozh river (near Tymchenky village), 24—Mozh river (near Fedorivka village).

One sampling site was selected for every water body within one ecoregion—the Eastern plains (code 16) [28] with different size rivers, such as small (S), medium (M), large (L) and extra-large (XL), passing through the underlying siliceous (Silicium, Si) or calcareous (Ca) rocks, are located in the lowland (1, below 200 m a.s.l.).

At each sampling site, the physicochemical parameters of water (Temperature, pH, Dissolved oxygen, Conductivity, Salinity and Total Dissolved Solids (TDS)) were measured using multifunction device AZ-86031; the results are presented in Table 1. A number of samples were taken for a further analysis, such as hydrochemistry, phytoplankton, phytobenthos, macrophytes, macroinvertebrates and fishes. The detailed analysis of biological data will be published soon.

**Table 1.** The physicochemical parameters of water and GIS coordinates of sampling sites in Siversky Donets river basin (Ukrainian part) in July–August of 2019.

| Site | Coordinates | | Type of Waterbody | Water Temperature, °C | pH | Dissolved Oxygen (DO), mg·L$^{-1}$ | DO, % | Conductivity, mS cm$^{-1}$ | Salinity, g·L$^{-1}$ (ppt) | Total Dissolved Solids, ppm |
|------|-------------|---|-------------------|------------------------|-----|-----------------------------------|-------|----------------------------|-----------------------------|------------------------------|
| 1 | N 49°58′16.4″ | E 37°51′07.8″ | UA_R_16_XL_1_Si | 26.0 | 6.53 | 5.7 | 67.4 | 749 | 0.44 | 375 |
| 2 | N 49°16′15.6″ | E 39°35′16.4″ | UA_R_16_M_1_Ca | 23.8 | 6.18 | 6.8 | 80 | 2240 | 1.18 | 1621 |
| 3 | N 49°35′27.4″ | E 38°12′08.5″ | UA_R_16_L_1_Si | 23.0 | 6.29 | 5.9 | 69 | 1388 | 0.87 | 696 |
| 4 | N 49°32′52.4″ | E 38°09′34.2″ | UA_R_16_M_1_Si | 23.2 | 6.2 | 7.2 | 74.6 | 2000 | 1.04 | 763 |
| 5 | N 48°52′55.3″ | E 37°54′04.3″ | UA_R_16_XL_1_Si | 23.6 | 6.48 | 6.5 | 83.2 | 1292 | 0.88 | 649 |
| 6 | N 48°58′19.6″ | E 37°58′51.8″ | UA_R_16_M_1_Si | 23.0 | 6.47 | 6.5 | 74 | 2380 | 1.26 | 1190 |
| 7 | N 48°25′46.4″ | E 37°40′31.5″ | UA_R_16_M_1_Si | 26.5 | 6.07 | 6.3 | 74.1 | 4800 | 2.65 | 2410 |
| 8 | N 49°53′09.5″ | E 36°43′18.5″ | UA_R_16_S_1_Si | 25.9 | 6.28 | 4.9 | 61.9 | 782 | 0.47 | 392 |
| 9 | N 49°47′36.1″ | E 36°19′49.2″ | UA_R_16_M_1_Si | 24.5 | 6.22 | 7.6 | 81 | 1313 | 0.8 | 657 |
| 10 | N 48°25′21.4″ | E 37°40′47.5″ | UA_R_16_M_1_Si | 20.8 | 7.88 | 14.3 | 155 | 5660 | 3.17 | 2830 |
| 11 | N 48°57′59.0″ | E 37°36′53.0″ | UA_R_16_XL_1_Si | 22.8 | 8.06 | 7.0 | 85 | 832 | 0.51 | 416 |
| 12 | N 48°55′10.9″ | E 39°37′45.1″ | UA_R_16_S_1_Ca | 13.8 | 7.75 | 11.6 | 117 | 2590 | 1.37 | 1290 |
| 13 | N 48°52′40.9″ | E 39°49′26.3″ | UA_R_16_L_1_Si | 18.2 | 7.85 | 10.1 | 108 | 2390 | 1.27 | 1190 |
| 14 | N 49°07′35.9″ | E 39°36′29.0″ | UA_R_16_L_1_Ca | 19.0 | 7.96 | 11.3 | 130 | 2070 | 1.09 | 1040 |
| 15 | N 49°17′20.2″ | E 40°06′22.1″ | UA_R_16_S_1_Ca | 12.0 | 7.68 | 7.5 | 68.4 | 2640 | 1.42 | 1330 |
| 16 | N 49°21′00.9″ | E 40°04′13.4″ | UA_R_16_S_1_Ca | 14.5 | 7.42 | 4.2 | 43 | 3170 | 1.72 | 1590 |
| 17 | N 49°22′32.1″ | E 38°55′57.3″ | UA_R_16_L_1_Si | 21.3 | 7.95 | 9.0 | 102 | 2040 | 1.08 | 1020 |
| 18 | N 48°59′40.2″ | E 38°29′25.1″ | UA_R_16_L_1_Si | 16.5 | 7.96 | 11.1 | 118 | 2250 | 1.2 | 1130 |
| 19 | N 48°49′57.5″ | E 38°21′18.6″ | UA_R_16_S_1_Ca | 16.5 | 7.91 | 9.8 | 93 | 2700 | 1.45 | 1350 |
| 20 | N 49°37′12.3″ | E 37°00′29.3″ | UA_R_16_M_1_Si | 16.4 | 7.68 | 7.4 | 84.3 | 2020 | 1.07 | 1010 |
| 21 | N 49°55′27.7″ | E 36°47′48.9″ | UA_R_16_M_1_Si | 16.3 | 7.73 | 5.8 | 60.4 | 980 | 0.59 | 485 |
| 22 | N 49°40′58.8″ | E 36°25′26.7″ | UA_R_16_XL_1_Si | 22.9 | 7.56 | 4.7 | 52.7 | 877 | 0.45 | 439 |
| 23 | N 49°44′37.8″ | E 36°09′04.5″ | UA_R_16_L_1_Si | 22.0 | 7.95 | 4.3 | 51.5 | 880 | 0.54 | 441 |
| 24 | N 49°48′45.0″ | E 35°44′44.0″ | UA_R_16_M_1_Si | 15.8 | 7.56 | 2.9 | 30.3 | 892 | 0.56 | 448 |

For hydrochemical studies, water samples were taken from the surface layer (~0.5 m) using a glass bathometer. The sample from the bathometer without filtration was poured into the plastic container for determination of the ion concentration in the laboratory of the Institute of Hydrobiology of the National Academy of Sciences of Ukraine according to standard methods [29].

A sample of 1.0–1.5 L$^{-1}$ was passed through the nitrocellulose filter (Synpor, Pragochem, Czech Republic) with a pore diameter of 0.4 μm to separate the suspended solids using the compressor unit 40-2M (M-Apparatura, Kharkiv, Ukraine). Removal of suspended solids is necessary to obtain reliable results. The concentration of inorganic forms of nitrogen and phosphorus, the values of permanganate index (CODMn) and the Dichromate Chemical Oxygen Demand (COD)) were determined in the obtained filtrate. The concentration of inorganic forms of nitrogen and phosphorus in water was determined using photometric techniques [29].

The samples of phytoplankton were collected by a bathometer at a depth of 0.5 m in all sampling sites processed in the laboratory of the Institute of Hydrobiology, determining the species composition of algae, as well as the quantitative characteristics of phytoplankton. Light microscopic (LM) observations were performed by means of Axio Imager A1 (Carl Zeiss, Oberkochen, Germany) light microscope with 40× HCX PLAN objective and 100× oil immersion objective lens (total magnification was 400–1000) equipped with digital camera AxioCam 506 color. Identification was performed using Süßwasserflora von Mitteleuropa [30–35], with some newer updates from Diatoms of Europe [36–38] and some additional monographs [39–48]. The identified taxa, as well as all lists of algal species from

previous studies of the territory, were validated using the AlgaeBase system [49] and "Algae of Ukraine . . . " monographic series [50–53]. The quantitative characteristics of algae was recorded by direct counting in a Nageotte chamber (volume 0.02 mL) using a light microscope Axio Imager A1 (Carl Zeiss, Oberkochen, Germany). The biomass of algae was obtained equating the cells to specific geometrical forms according to Hillebrand et al. [54].

### 2.3. Ecological Analysis Based on Species Composition and Quantitative Characteristics of Phytoplankton

Each group of algae identified in the Siversky Donets river basin responds to environmental variables and can be used as an indicator reflecting the response of aquatic ecosystems to habitat preference, flow and oxygenation, pH, salinity, trophic state and class of organic pollution [55]. As for habitat preference, the species found in the studied river basin are divided into the following groups: planktonic (P), plankto-benthic (P-B), benthic (B), epiphyte (Ep), soil (S) and phycobiont (pb). The temperature preferences of identified algae were divided as follows: cool water (cool), temperate (temp), eurythermic (eterm) and warm water (warm). The flow and oxygenation regime characterize the following groups of indicators: standing water (st), streaming water (str), low streaming water (st-str) and aerophiles (ae). The pH indicators were presented by alkalibiontes (alb), alkaliphiles (alf), indifferents (ind) and acidophiles (acf). Salinity indicator groups in the Siversky Donets river basin were the following: oligohalobes-halophobes (hb), oligohalobes-indifferents (i), mesohalobes (mh) and halophiles (hl). Trophic state indicators in the studied basin were divided as follows: oligotraphentic (ot), oligo-mesotraphentic (o-m), mesotraphentic (m), meso-eutraphentic (me), eutraphentic (e), hypereutraphentic (he) and oligo- to eutraphentic (hypereutraphentic) (o-e). Nitrogen uptake metabolism indicators were presented by nitrogenautotrophic taxa (ats), tolerating very small concentrations of organically bound nitrogen; nitrogen-autotrophic taxa (ate), tolerating elevated concentrations of organically bound nitrogen; facultatively nitrogen-heterotrophic taxa (hne), needing periodically elevated concentrations of organically bound nitrogen; obligatory nitrogen-heterotrophic taxa (hce), needing continuously elevated concentrations of organically bound nitrogen. Organic pollution indicator groups according to Watanabe (diatoms only) in the Siversky Donets river basin were divided into saproxenes (sx), saprophils (sp) and eurysaprobes (es).

Identification of organic pollution was performed by estimating the values of saprobity indices and indicator groups, followed by assigning them to corresponding water quality classes. [56]. Considering identified indicator groups, the intervals of the water quality classes calculated on the basis of organic pollution by Sládeček are distributed as follows: I (high class of water quality)—0–0.5 (xenosaprobionts, xeno-oligosaprobionts), II (good)—0.6–1.5 (oligo-xenosaprobionts, xeno-beta-mesosaprobionts, oligosaprobionts, oligo-beta-mesosaprobionts), III (moderate)—(beta-oligosaprobionts, oligo-alpha-mesosaprobionts, beta-mesosaprobionts, beta-alpha-mesosaprobionts) and IV (bad)—(alpha- mesosaprobionts) [17,20,55,57,58].

### 2.4. Statistical Data Analysis

We used R package 'vegan' for the identification of the essential environmental variables affecting phytoplankton abundance and community structure, by constrained ordination [59,60]. All phytoplankton and environmental data were rank-transformed to eliminate the distribution unevenness and reduce the effect of outliers.

The environmental variables that drive heterogeneity of phytoplankton community structure were identified. According to the suggestion of Šmilauer and Lepš [61], detrending correspondence analysis (DCA) was used first to find whether phytoplankton abundance data showed linear or unimodal responses to the underlying gradients. Because the lengths of gradient were more than 4, we conducted the canonical correspondence analysis (CCA) [62]. Explanatory environmental variables were chosen by the stepwise selection procedure based on Monte-Carlo permutation tests of the constraint's significance implemented in the ordistep function, and only those variables that were significantly related to community structure ($p < 0.05$) were selected to be considered in CCA, and to

be shown in the ordination diagram, as suggested by Legendre and Legendre [59]. In the present analysis, Permanganate Index (CODMn), TDS, $NO_2^-$, $HCO_3^-$, COD and $Mg^{2+}$ were included as environmental variables that significantly influenced phytoplankton community. To improve the estimation of the effect of $HCO_3^-$, the effect of $Mg^{2+}$ was excluded. As the community structure parameters, Saprobic Index (SI), the total number of species (Nsp), species number per divisions of Bacillariophyta, Chlorophyta and Cyanobacteria and family's number were included in the model.

## 3. Results

### 3.1. Physicochemical Characteristics of the Sampling Sites

Hydrochemical composition is known as one of the most important environmental factors that significantly affect the productivity, growth, reproduction, stability, physiological and biochemical processes of aquatic organisms. In addition, according to the EU Water Framework Directive [14], physicochemical parameters are one of the three main elements for establishing the environmental status (potential) of the water bodies.

Within this study, we measured and analyzed the physicochemical parameters on sites of the Ukrainian part of the Siversky Donets river basin (Tables 1 and 2).

**Table 2.** The hydrochemical characteristics of water in sampling sites of the Siversky Donets river basin (Ukrainian part) in July–August of 2019.

| | Survey Area | $N\text{-}NH_4^+$, $mg\cdot L^{-1}$ | $N\text{-}NO_2^-$, $mg\cdot L^{-1}$ | $N\text{-}NO_3^-$, $mg\cdot L^{-1}$ | $P\text{-}PO_4^{3-}$, $mg\cdot L^{-1}$ | $COD_{Mn}$, $mg\cdot L^{-1}$ | COD, mg $O\cdot L^{-1}$ | Hardness, $mmol\cdot L^{-1}$ | $Ca^{2+}$, $mg\cdot L^{-1}$ | $Mg^{2+}$, $mg\cdot L^{-1}$ | $Cl^-$, $mg\cdot L^{-1}$ | $SO_4^{2-}$, $mg\cdot L^{-1}$ | $HCO_3^-$, $mg\cdot L^{-1}$ | $K^+ + Na^+$, $mg\cdot L^{-1}$ |
|---|---|---|---|---|---|---|---|---|---|---|---|---|---|---|
| 1 | Oskil river | 0.220 | 0.025 | 0.707 | 0.510 | 7.17 | 16.00 | 6.6 | 104.21 | 17.01 | 53.18 | 196 | 292.8 | 94.50 |
| 2 | Derkul river (near Bilovodsk village) | 0.112 | 0.011 | 0.258 | 0.090 | 9.12 | 56.00 | 13.6 | 190.38 | 49.82 | 350.96 | 288 | 298.9 | 180.00 |
| 3 | Krasna river (near Nyzhnya Duvanka village) | 0.145 | 0.011 | 0.257 | 0.204 | 9.45 | 41.60 | 9.6 | 128.26 | 38.88 | 148.89 | 260 | 341.6 | 140.50 |
| 4 | Krasna river (near Preobrazhennya village) | 0.246 | 0.002 | 0.043 | 0.210 | 9.78 | 54.40 | 12.2 | 160.32 | 51.03 | 209.16 | 296 | 366.0 | 146.75 |
| 5 | Siversky Donets river (near Kryva Luka village) | 0.163 | 0.025 | 0.212 | 0.473 | 10.11 | 35.20 | 8.6 | 112.22 | 36.45 | 159.53 | 224 | 268.4 | 124.25 |
| 6 | Zherebets river | 0.146 | 0.015 | 0.378 | 0.200 | 9.78 | 51.20 | 12.6 | 148.3 | 63.18 | 202.06 | 280 | 311.1 | 100.75 |
| 7 | Bychok river | 0.351 | 0.002 | 0.052 | 0.077 | 18.92 | 102.40 | 26.0 | 308.62 | 128.79 | 478.58 | 206 | 427.0 | 132.00 |
| 8 | Tetlyha river | 0.208 | 0.001 | 0.242 | 0.135 | 8.47 | 16.00 | 6.0 | 84.17 | 21.87 | 265.88 | 104 | 341.6 | 231.75 |
| 9 | Udy river | 14.725 | 0.761 | 0.184 | 2.292 | 14.35 | 22.40 | 7.6 | 104.21 | 29.16 | 88.62 | 32 | 372.1 | 145.75 |
| 10 | Bychok river | 0.220 | 0.002 | 0.046 | 0.101 | 14.08 | 188.77 | 30.0 | 360.72 | 145.8 | 808.26 | 112 | 427.0 | 53.25 |
| 11 | Siversky Donets river (near Mayaky village) | 0.259 | 0.015 | 0.544 | 0.326 | 9.60 | 20.64 | 5.7 | 64.13 | 30.38 | 93.94 | 164 | 219.6 | 99.25 |
| 12 | Chuhynka river | 0.077 | 0.016 | 3.561 | 0.182 | 5.76 | 44.24 | 16 | 256.51 | 38.88 | 152.44 | 348 | 384.3 | 46.25 |
| 13 | Derkul river (near Krasnyy Derkul village) | 0.158 | 0.003 | 0.050 | 0.189 | 10.88 | 94.37 | 15.2 | 148.3 | 94.77 | 421.86 | 252 | 378.2 | 203.75 |
| 14 | Derkul river (near Novoderkul village) | 0.079 | 0.008 | 0.558 | 0.053 | 6.72 | 97.32 | 13.4 | 204.41 | 38.88 | 386.41 | 228 | 335.5 | 193.75 |
| 15 | Cherepakha river | 0.145 | 0.013 | 1.164 | 0.127 | 9.28 | 123.88 | 18.5 | 286.57 | 51.03 | 584.93 | 180 | 359.9 | 191.25 |
| 16 | Milova river | 0.258 | 0.002 | 0.050 | 0.100 | 13.76 | 123.88 | 22.5 | 416.83 | 20.66 | 553.02 | 248 | 475.8 | 151.75 |
| 17 | Aidar river | 0.123 | 0.002 | 0.064 | 0.126 | 9.60 | 23.6 | 12.9 | 112.22 | 88.7 | 322.6 | 252 | 335.5 | 173.75 |
| 18 | Borova river | 0.059 | 0.007 | 0.634 | 0.096 | 7.36 | 47.19 | 14.8 | 210.42 | 52.25 | 265.88 | 308 | 317.2 | 108.00 |
| 19 | Bilenka river | 0.053 | 0.002 | 1.458 | 0.023 | 6.08 | 53.08 | 15.5 | 214.43 | 58.32 | 210.93 | 268 | 420.9 | 73.25 |
| 20 | Serednia Balakliika river | 0.140 | 0.002 | 0.147 | 0.100 | 8.96 | 23.59 | 9.5 | 80.16 | 66.83 | 53.18 | 264 | 555.1 | 165.00 |
| 21 | Velyka Babka river | 0.067 | 0.002 | 0.074 | 0.249 | 9.60 | 20.64 | 7.6 | 112.22 | 24.30 | 42.54 | 212 | 396.5 | 113.00 |
| 22 | Siversky Donets river (near Cheremushne village) | 4.685 | 0.355 | 1.665 | 0.984 | 10.88 | 29.49 | 5.8 | 74.15 | 25.52 | 81.54 | 140 | 317.2 | 115.50 |
| 23 | Mozh river (near Tymchenky village) | 0.274 | 0.004 | 0.049 | 0.147 | 10.88 | 29.49 | 7.7 | 98.2 | 34.02 | 60.26 | 52 | 515.5 | 88.25 |
| 24 | Mozh river (near Fedorivka village) | 0.252 | 0.005 | 0.059 | 0.624 | 14.4 | 41.29 | 7.4 | 90.18 | 35.24 | 63.81 | 50 | 509.4 | 94.75 |

The water temperature in the sampling sites varied in a wide range of +12.0–+26.5 °C, caused by different hydrological regimes in the studied watercourses. The pH was in the range of slightly acidic to slightly basic, 6.07–8.06.

Total mineralization greatly varied, from 375 to 2830 ppm. The lowest values were observed in the Oskil, Tetlega and Siversky Donets rivers, and the highest in the Derkul (left tributary of the Seversky Donets basin) and Bychok (right tributary of the Seversky Donets basin) rivers (Table 1). $Ca^{2+}$ dominated the cationic composition in most watercourses, its content varied from 64.13 to 416.83 $mg\cdot L^{-1}$. Together with $Mg^{2+}$, $Ca^{2+}$ determines water hardness. The surface waters of the Siversky Donets basin in this study were from moderately hard to hard.

The content of dissolved oxygen greatly varied, in the range from 2.9 to 14.3 $mg\cdot L^{-1}$. The highest values were observed in the Bychok, Siversky Donets and Derkul, while the lowest values were observed in the Mzha and Milova rivers.

The permanganate oxidation ($COD_{Mn}$) differed in the range from 5.76 to 18.92 mg·$L^{-1}$. Its largest values were observed in the Bychok, Mzha and Uda rivers.

Within current study, we identified the following values of chemical oxygen demand (COD): the maximum was recorded in the rivers Bychok (188.77 mg O·$L^{-1}$) and Milova and Cherepakha (123.88 mg O·$L^{-1}$), while quite high values were recorded in the river Derkul in the village Novoderkul and Krasnyy Derkul (97.32 and 94.37 mg O·$L^{-1}$, respectively). The moderate values of COD (35–60 mg O·$L^{-1}$) were typical for the rivers Zherebets, Krasna, Derkul, Chuhynka, Borova, Bilenka, Mozh and Siversky Donets (Donetsk region). Other rivers were characterized by low COD values, which were within the range of 16–30 mg O·$L^{-1}$.

In most studied watercourses, the concentration of dissolved ammonium, the most easily uptook nitrogen source for phytoplankton, ranged within 0.053–0.351 mg N·$L^{-1}$, whereas in the Siversky Donets (Kharkiv region, Cheremushne village) and the Udy rivers its concentration was at least by an order higher than average.

Significantly lower compared to the Udy river, but fairly high concentrations of ammonium were observed in the rivers Mozh, Milova and Siversky Donets (Donetsk region). Moderate levels of ammonium concentration were registered in the rivers Zherebets, Siversky Donets, Krasna (downstream the village of Nyzhnia Duvanka), Serednaia Balakliika, Aidar, Cherepakha and Derkul (village of Novoderkul), and minimum values were measured in the rivers Bilenka, Borova, Velyka Babka, Derkul (village of Krasnyy Derkul) and Chuhynka.

Nitrite concentrations in the study were characterized by low values (0.001–0.025 mg N·$L^{-1}$) (see Table 2). However, they were relatively high in the Siversky Donets river (Cheremushne) and the Udy river, correlating with high concentrations of ammonium.

The concentration of nitrate ions was almost twice higher than the average in the Cherepakha, Bilenka and Siversky Donets (Kharkiv region) rivers. The maximum concentration of nitrate ions was observed in the Chuhynka River, where it was 3.551 mg N·$L^{-1}$, which is seven times higher than the average concentration, 0.519 mg N·$L^{-1}$.

Among the surveyed sections of the rivers of the Siversky Donets basin in July, the maximum concentration of orthophosphate ions was observed in the waters of the Udy river (2.292 mg P·$L^{-1}$, see Table 2). In the other rivers, it was significantly lower and varied in the range of 0.077–0.500 mg P·$L^{-1}$ (see Table 2). Its minimum was recorded in the rivers Bychok (0.077 mg P·$L^{-1}$) and Derkul (0.09 mg P·$L^{-1}$), the average concentration of about 0.2 mg·$L^{-1}$ was typical for the rivers Tetlyha, Stallion and Krasna, while in the rivers Oskil and Siversky Donets it was 2.5 times higher than average.

Measurements in August–September showed that in the most basin watercourses, the concentration of orthophosphate ions was also low and fluctuated within the range of 0.005–0.2 mg·$L^{-1}$, which is typical for unpolluted natural waters. The exceptions were the rivers Velyka Babka (0.249 mg P·$L^{-1}$), Siversky Donets (Donetsk region) (0.326 mg P·$L^{-1}$) and Mozh (the village of Fedorivka) (0.624 mg P·$L^{-1}$). A maximum value of 0.984 mg P·$L^{-1}$ was registered in the Siversky Donets river (Kharkiv region), which was more than twice as small compared to the Udy river (2.292 mg P·$L^{-1}$) (Table 2). Overall, the highest concentration of orthophosphate ions was common in the water bodies within the Kharkiv region.

### 3.2. Species Composition of Algae in Sampling Sites of the Siversky Donets River Basin

The species composition of the algae in the Sirevsky Donets river basin comprised 167 species of algae (168 intraspecies taxa) mostly formed by Bacillariophyta (63%) and Chlorophyta (22%) and some other groups (Figure 2). Thus, the diversity was formed from Bacillariophyta—103 species (104 intfraspecies taxa), Chlorophyta—37 species, Cyanobacteria—10, Charophyta—5, Chrysophyta—2, Dinophyta—2 and Euglenophyta—7 (Tables 3 and S1).

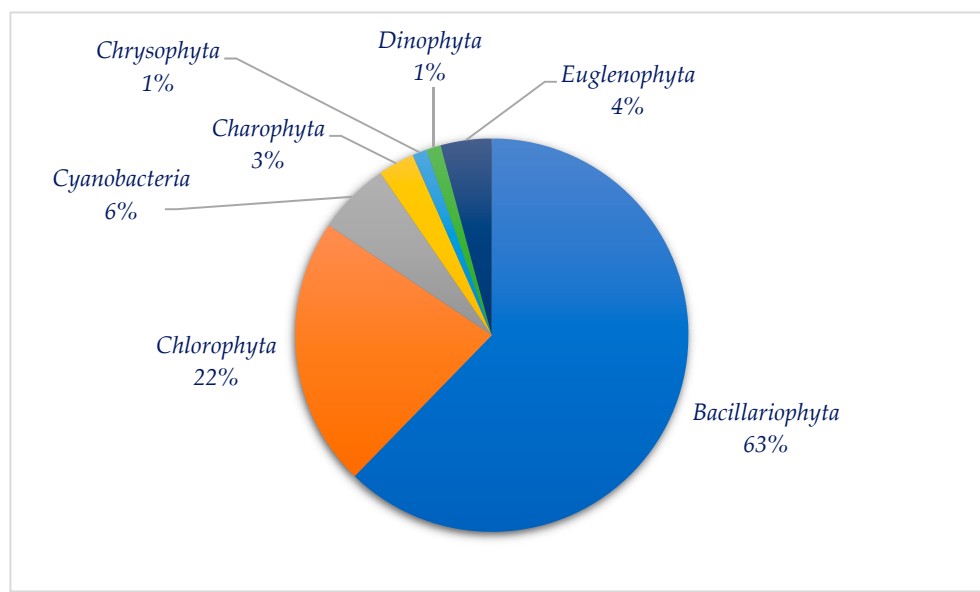

**Figure 2.** Taxonomic spectrum of phytoplankton in the Siversky Donets river basin (Ukrainian part).

**Table 3.** Taxonomic composition of phytoplankton in the Siversky Donets river basin.

| | Sites | | | | | | | | | | | | | | | | | | | | | | | |
|---|---|---|---|---|---|---|---|---|---|---|---|---|---|---|---|---|---|---|---|---|---|---|---|---|
| | 1 | 2 | 3 | 4 | 5 | 6 | 7 | 8 | 9 | 10 | 11 | 12 | 13 | 14 | 15 | 16 | 17 | 18 | 19 | 20 | 21 | 22 | 23 | 24 |
| Phylum (Division) | 3 | 4 | 4 | 3 | 5 | 4 | 6 | 3 | 5 | 3 | 3 | 3 | 2 | 1 | 2 | 1 | 3 | 3 | 3 | 3 | 1 | 4 | 4 | 6 |
| Class | 6 | 8 | 7 | 5 | 7 | 6 | 7 | 4 | 9 | 4 | 5 | 3 | 3 | 2 | 3 | 2 | 4 | 4 | 4 | 5 | 2 | 7 | 5 | 7 |
| Order | 11 | 14 | 10 | 10 | 13 | 9 | 10 | 7 | 12 | 9 | 10 | 8 | 8 | 7 | 7 | 5 | 12 | 8 | 9 | 7 | 5 | 15 | 8 | 8 |
| Family | 14 | 15 | 11 | 10 | 16 | 12 | 11 | 8 | 13 | 12 | 12 | 8 | 9 | 8 | 11 | 6 | 12 | 10 | 9 | 8 | 8 | 20 | 10 | 9 |
| Species | 24 | 19(20) | 19(20) | 14(15) | 25 | 21 | 14 | 16 | 20 | 18 | 19 | 12 | 10 | 10 | 12 | 9 | 12 | 11 | 13(14) | 10 | 10 | 35 | 11 | 13 |

Studied water bodies have been analyzed using significant parameters of phytoplankton composition, which are used for the ecological status specification [14] with a purpose to use these data for the future monitoring investigations in the Siversky Donets river basin. The number of species varied from maximum in the Siversky Donets river (near Christine village) (UA_R_16_XL_1_Si), 35 species, to the minimum, 9 species, in the Milova river (UA_R_16_S_1_Ca). The highest and the lowest number of families is noted for the same sites (see Table 3).

The sites merged together in two groups according to the river's sizes: small (S) + medium (M)—2, 3, 6, 7, 8, 9, 10, 12, 15, 16, 19, 20, 21, 24 and large (L) + extra-large (XL)—1, 4, 5, 11, 13, 14, 17, 18, 22, 23, which revealed similar features among the group and between different groups. As can be noted in Figure 3, the higher divisions number as well as species correspond to the rivers with smaller size, while bigger size rivers possessed the lesser divisions' numbering.

Among the Ukrainian rare species, the following were identified in the Siversky Donets river basin: *Humidophila perpusilla* (Grunow) Lowe, Kociolek, J.R. Johansen et al. (=*Diadesmis perpusilla* (Grunow) D.G. Mann), which detected in phytoplankton from the river Zherebets and *Eunotia siberica* Cleve, found in the river Udy. In addition, *Phacus snitkovii* Roll was identified in the plankton of the river Zherebets. This taxon was described from the territory of Ukraine, but it was only the second finding of this species for the river Siversky Donets; the first was made by Svirenko as early as 1938 [63].

Quantitative parameters of phytoplankton (abundance and biomass) were apportioned unevenly. As can be observed in Figure 4, the maximum abundance was on sites 9 (Udy river), 11 (Siversky Donets river (near Mayaky village) and 17 (Aidar river) presented by 13,370, 20,265 and 60,111 th. cells·L$^{-1}$, respectively. In spite of high abundance, the biomass values were not that high with the maximum values on sites 5 (Siversky Donets river (near Kryva Luka village), 9 (Udy river) and 11 (Siversky Donets river (near

Mayaky village). The overall low ratio of abundance to biomass is indicative of an increased proportion of small-celled algae.

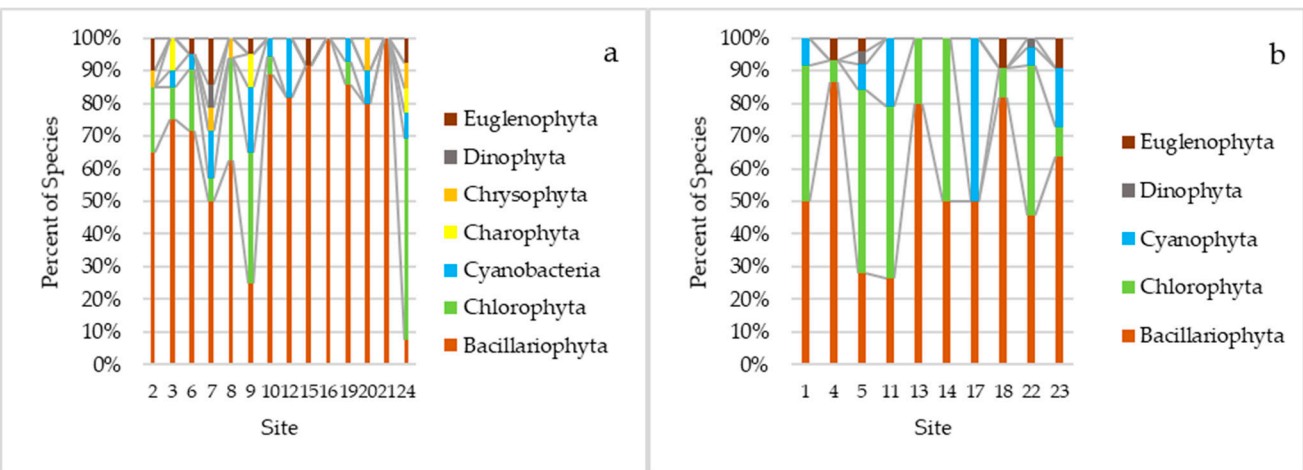

**Figure 3.** Divisions of phytoplankton grouped according to size of rivers: S and M (**a**); L and XL (**b**) in the Siversky Donets river basin (Ukrainian part).

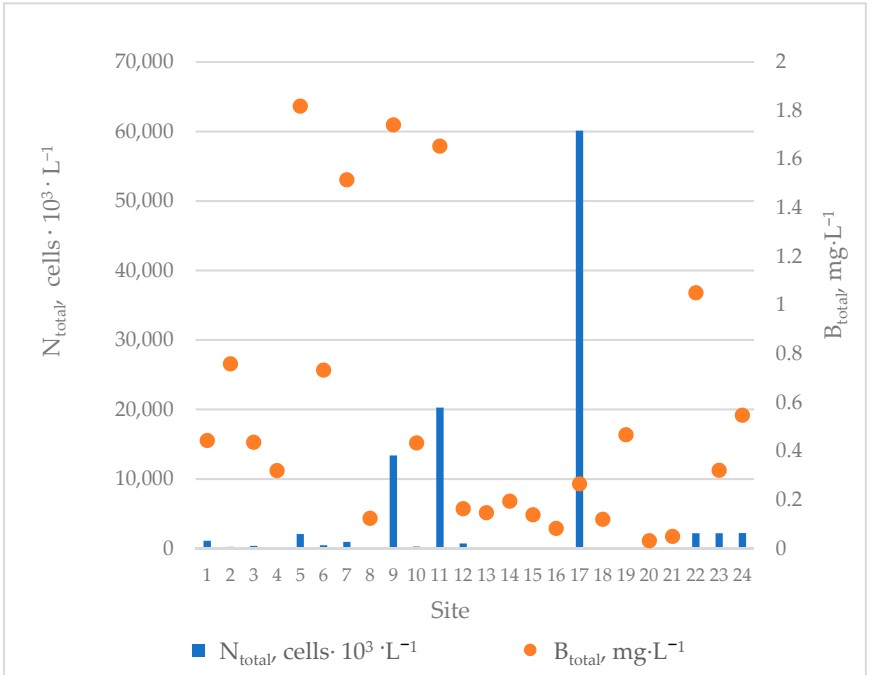

**Figure 4.** Abundance (N) and biomass (B) of phytoplankton in the Siversky Donets river basin (Ukrainian part).

A representative integral water parameter indicative of phytoplankton biomass and water quality is chlorophyll $a$ concentration [14]. According to published data, the concentration of chlorophyll $a$ in the studied rivers in the summer was in the range of $\geq 6\ \mu g \cdot L^{-1}$ [27], which corresponds to oligotrophic and mesotrophic waters. The exceptions are the Uda and Aidar watercourses, where the concentration was more than $20\ \mu g \cdot L^{-1}$. An increased chlorophyll $a$ concentration indicates an increase in the functional activity of phytoplankton and the trophic level of water, which is consistent with the high phytoplankton abundance and the concentration of the main nutrients observed in our study.

To study groups that formed high values of the quantitative parameters, Figures 5 and 6 are presented in this paper. The maximal abundance was explained by the high values of Cyanobacteria in site 9 (11,100 th. cells·L$^{-1}$), 11 (19,050 th. cells·L$^{-1}$) and 17 (60,000 th. cells·L$^{-1}$). Considering the correspondence of the cells' abundance to biomass values, the water "bloom" did not occur in these sites. The high biomass values on the noted above sites (5, 9, 11) was predominantly formed by Dinophyta (0.86 mg·L$^{-1}$), Chlorophyta (0.45 mg·L$^{-1}$) and Bacillariophyta (0.41 mg·L$^{-1}$) on site 5, by Cyanobacteria (0.87 mg·L$^{-1}$) and Bacillariophyta (0.72 mg·L$^{-1}$) on site 9 and by Cyanobacteria (1.24 mg·L$^{-1}$) on site 11 (Figure 5). The abundance data (cells' number) for each site revealed heterogeneous dominant species (Table 4); the same was prevalent in biomass (Table 5).

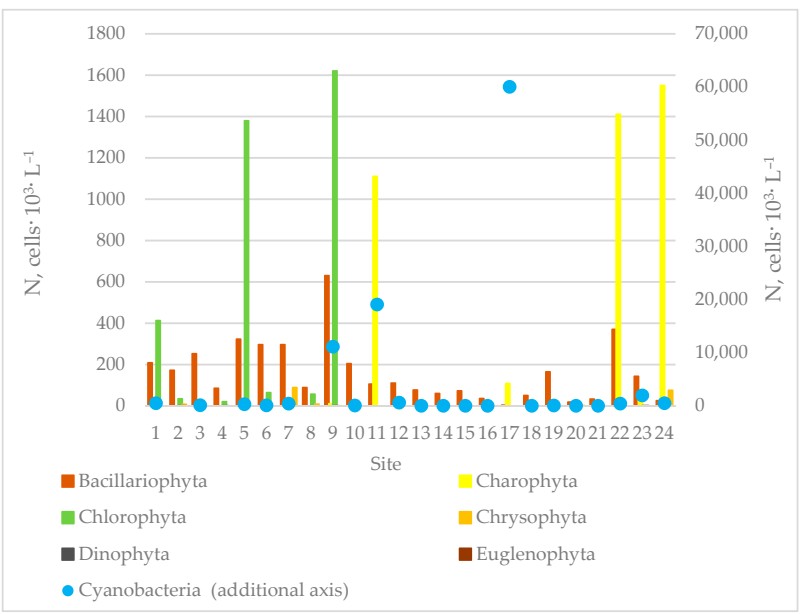

**Figure 5.** The abundance of the phytoplankton cells by divisions within the Ukrainian part of the Siversky Donets river basin. For a better visibility, an additional axis was added referring to Cyanobacteria.

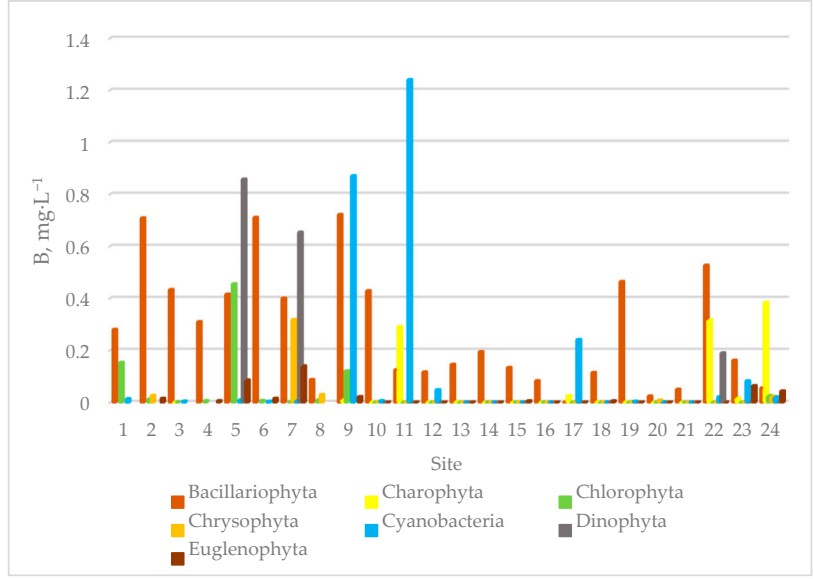

**Figure 6.** Biomass of phytoplankton cells in divisions within the Ukrainian part of the Siversky Donets river basin.

**Table 4.** Dominant species of algae according to abundance values in the river basin of Siversky Donets.

| Species | Sites' Number |
|---|---|
| *Anagnostidinema amphibium* (C. Agardh ex Gomont) Strunecký, Bohunická, J.R. Johansen & J. Komárek | 11 |
| *Aphanizomenon flosaquae* Ralfs ex Bornet & Flahault | 9, 11, 12 |
| *Aphanocapsa planctonica* (G.M. Smith) Komárek & Anagnostidis | 17 |
| *Aulacoseira granulata* (Ehrenberg) Simonsen | 13, 15, 16, 18–20 |
| *Cocconeis pediculus* Ehrenberg | 2, 4, 6 |
| *Cocconeis placentula* Ehrenberg | 13, 14, 18 |
| *Coelastrum pseudomicroporum* Korshikov | 24 |
| *Cyclotella meneghiniana* Kützing | 3, 5, 7, 10 |
| *Desmodesmus communis* (E. Hegewald) E. Hegewald | 4 |
| *Desmodesmus costato-granulatus* (Skuja) E. Hegewald | 8, 22 |
| *Dolichospermum flosaquae* (Brébisson ex Bornet & Flahault) P. Wacklin, L. Hoffmann & J. Komárek | 10, 23 |
| *Gomphonema constrictum* Ehrenberg | 10 |
| *Halamphora veneta* (Kützing) Levkov | 14 |
| *Lemmermannia triangularis* (Chodat) C. Bock & Krienitz | 5 |
| *Melosira varians* C. Agardh | 2 |
| *Microcystis viridis* (A. Braun) Lemmermann | 5 |
| *Nitzschia vermicularis* (Kützing) Hantzsch | 8 |
| *Oscillatoria limosa* C. Agardh ex Gomont | 22, 24 |
| *Oscillatoria planctonica* Woloszynska | 1, 6 |
| *Oscillatoria tenuis* C. Agardh ex Gomont | 1, 7, 11, 12, 19, 20 |
| *Oscillatoria ucrainica* Vladimirova | 23 |
| *Phormidium terebriforme* (C. Agardh ex Gomont) Anagnostidis & Komárek | 3 |
| *Raphidocelis sigmoidea* Hindák | 8 |
| *Rhoicosphenia abbreviata* (C. Agardh) Lange-Bertalot | 21 |
| *Stephanodiscus hantzschii* Grunow | 21 |
| *Ulnaria ulna* (Nitzsch) Compère | 15 |
| *Willea apiculata* (Lemmermann) D.M. John, M.J. Wynne & P.M. Tsarenko | 1, 5 |

**Table 5.** Dominant species of algae according to biomass values in the Siversky Donets river basin.

| Species | Division | Sites' Number |
|---|---|---|
| *Amphora pediculus* (Kützing) Grunow | Bacillariophyta | 18 |
| *Aphanizomenon flosaquae* Ralfs ex Bornet & Flahault | Cyanobacteria | 9 |
| *Aphanocapsa planctonica* (G.M. Smith) Komárek & Anagnostidis | Cyanobacteria | 17 |
| *Aulacoseira granulata* (Ehrenberg) Simonsen | Bacillariophyta | 19, 20 |
| *Campylodiscus noricus* Ehrenberg ex Kützing | Bacillariophyta | 19 |
| *Cocconeis pediculus* Ehrenberg | Bacillariophyta | 2–4, 6, 10 |
| *Cocconeis placentula* Ehrenberg | Bacillariophyta | 1, 13, 18, 23, 24 |
| *Cocconeis placentula* var. *euglypta* (Ehrenberg) Grunow | Bacillariophyta | 6 |
| *Coelastrum pseudomicroporum* Korshikov | Chlorophyta | 24 |
| *Cyclotella meneghiniana* Kützing | Bacillariophyta | 5, 7, 10 |
| *Diatoma tenuis* C. Agardh | Bacillariophyta | 8 |
| *Gymnodinium paradoxum* A.J. Schilling | Dinophyta | 5 |
| *Halamphora veneta* (Kützing) Levkov | Bacillariophyta | 12, 14 |
| *Mallomonas* sp. | Ochrophyta | 7, 8, 20 |
| *Melosira varians* C. Agardh | Bacillariophyta | 2, 4 |
| *Navicula recens* (Lange-Bertalot) Lange-Bertalot | Bacillariophyta | 20 |
| *Navicula vulpina* Kützing | Bacillariophyta | 16 |
| *Nitzschia acicularis* (Kützing) W. Smith | Bacillariophyta | 12 |
| *Nitzschia vermicularis* (Kützing) Hantzsch | Bacillariophyta | 8 |
| *Oscillatoria tenuis* C. Agardh ex Gomont | Cyanobacteria | 11 |
| *Oscillatoria ucrainica* Vladimirova | Cyanobacteria | 23 |
| *Peridiniopsis quadridens* (F. Stein) Bourrelly | Dinophyta | 22 |
| *Rhoicosphenia abbreviata* (C. Agardh) Lange-Bertalot | Bacillariophyta | 21 |
| *Sphaerocystis planctonica* (Korshikov) Bourrelly | Chlorophyta | 1 |
| *Surirella librile* (Ehrenberg) Ehrenberg | Bacillariophyta | 14 |
| *Trachelomonas volvocina* (Ehrenberg) Ehrenberg | Euglenophyta | 23 |
| *Ulnaria acus* (Kützing) Aboal | Bacillariophyta | 13, 14, 16 |
| *Ulnaria biceps* (Kützing) Compère | Bacillariophyta | 15 |
| *Ulnaria ulna* (Nitzsch) Compère | Bacillariophyta | 10, 13, 15, 16, 18, 21 |
| *Willea apiculata* (Lemmermann) D.M. John, M.J. Wynne & P.M. Tsarenko | Chlorophyta | 24 |

### 3.3. Ecological Analysis Based on Species Composition and Indicator Characteristics of Phytoplankton

The indicator characteristics of the whole algal species list identified in plankton samples in the Ukrainian part of the Siversky Donets river basin included preferences regarding habitat conditions, temperature, pH, salinity, oxygenation, organic pollution according to Watanabe and Sládeček, trophic state and nitrogen uptake metabolism (autotrophy-heterotrophy) [55]. The indicators for the habitat preferences comprised 124 taxa: 70 plankto-benthic algae in a broad sense (P-B; P-B, Ep; P-B, S; P-B, Ep, S; P-B, pb, S), 37 benthic algae taxa (B; B, S; B, aer) and 17 strictly plankton species (P). The indicators of the temperature regime included 32 taxa that were divided into groups as follows: 23 temperate taxa (temp), 6 eurytherms (eterm), 2 cold water forms (cool) and 1 warm water form (warm). The number of indicators of water mass dynamics and oxygenation equaled 99 and was mostly formed by species indicating waters with moderate dynamics and oxygenation (limnophiles and rheo-limnophils) (st-str; st-str, $H_2S$)—71 taxa. Species preferring fast-moving highly oxygenated waters (rheobionts and rheophiles) (str) comprised 12 taxa, and species typical for almost standing poorly oxygenated waters (limnobionts) comprised 16 taxa. A total of 83 species were found that allowed characterizing the pH regime in the river basin, distributed as follows: 41 alkaliphiles (alf), 39 indifferents (ind), 2 alkalibionts (alb) and 1 acidophile (acf). The number of species characterizing water salinity was 105, formed mostly by 87 indifferents (i), 1 halophobes (hb), 14 halophiles (hl) and 3 mesohalobionts (mh). The indicators of the Watanabe's Organic pollution comprised 60 taxa: 39 eurysaprobic species (es), 15 saproxenous species (sx) and 6 saprophilous species (sp). The total number of the indicators of organic pollution according to Pantle-Buck in the Sládeček modification was formed by 120 taxa, with I class of water quality composed of 2 xenosaprobes (x) and 1 xeno-oligosaprobes (x-o), II class of water quality composed of 14 oligosaprobes (o), 14 oligo-betamesosaprobes (o-b), 5 oligo-xenosaprobes (o-x) and 2 xeno-betamesosaprobes (x-b), III class of water quality composed of 42 betamesosaprobes (b), 8 beta-alphamesosaprobes (b-a), 12 oligo-alphasaprobes (o-a) and 5 beta-oligosaprobes (b-o) and IV class of water quality composed of 12 oligo-alphasaprobes (a-o) and 3 alphamesosaprobes (a).

Trophic state indicators were formed by 77 taxa: 6 oligotraphentic (ot), 14 oligo-mesotraphentic (o-m), 7 mesotraphentic (m), 27 meso-eutraphentic (me), 16 eutraphentic (e), 2 hypereutraphetic (he) and 5 oligo-to-eutraphentic (o-e). Nitrogen uptake metabolism (photosynthetic activity) indicators included 54 taxa that were divided into groups as follows: 31 nitrogen-autotrophic taxa tolerating elevated concentrations of organically bound nitrogen (ate), 13 nitrogen-autotrophic taxa tolerating very small concentrations of organically bound nitrogen (ats), 5 obligatory nitrogenheterotrophic taxa needing continuously elevated concentrations of organically bound nitrogen (hce) and 5 facultative nitrogen-heterotrophic taxa needing periodically elevated concentrations of organically bound nitrogen (hne).

In turn, the ecological characteristics of phytoplankton was described for two main river sizes: small (S) + medium (M) and large (L) + extra-large (XL). The sites numbers 2, 3, 6, 7, 8, 9, 10, 12, 15, 16, 19, 20, 21 and 24 belong to the S + M river size (Figures 7a,c,e,g and 8a,c,e,g and the sites numbers 1, 4, 5, 11, 13, 14, 17, 18, 22 and 23 belong to the L + XL river size (Figure 7b,d,f,h and Figure 8b,d,f,h).

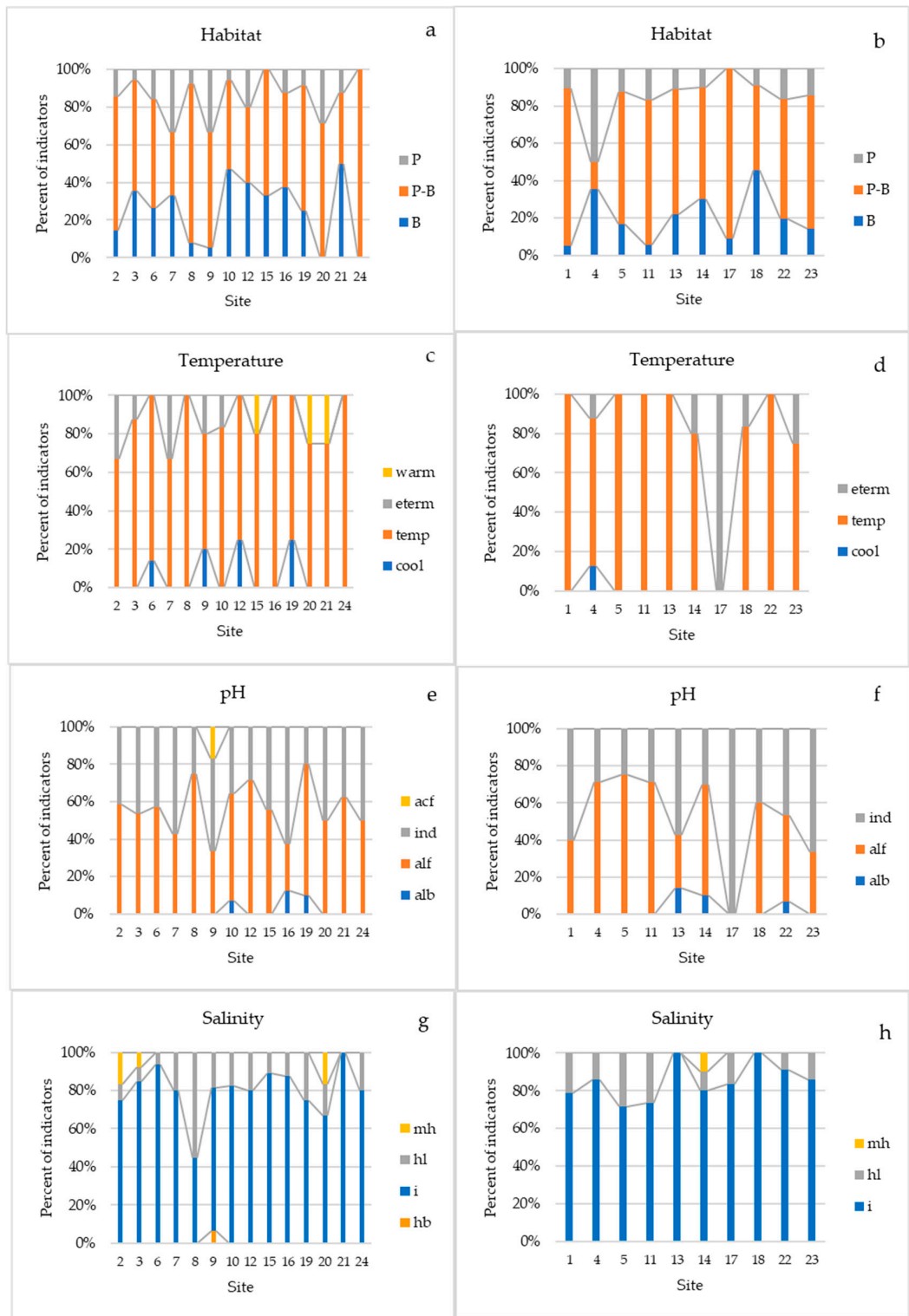

**Figure 7.** Ecological characteristics of phytoplankton species indicators (habitat, temperature, pH, salinity) grouped according to the size of rivers, S and M (**a**,**c**,**e**,**g**) and L and XL (**b**,**d**,**f**,**h**), within the Ukrainian part of the Siversky Donets river basin. Abbreviations of ecological groups: by habitat, P—planktonic, P-B—plankto-benthic, B—benthic; by temperature, cool—cool water, temp—temperate, eterm—eurythermic, and warm—warm water; by pH, alb—alkalibiontes, alf—alkaliphiles, ind—indifferents, and acf—acidophiles; by salinity, hb—oligohalobes-halophobes, i—oligohalobes-indifferents, mh—mesohalobes and hl—halophiles.

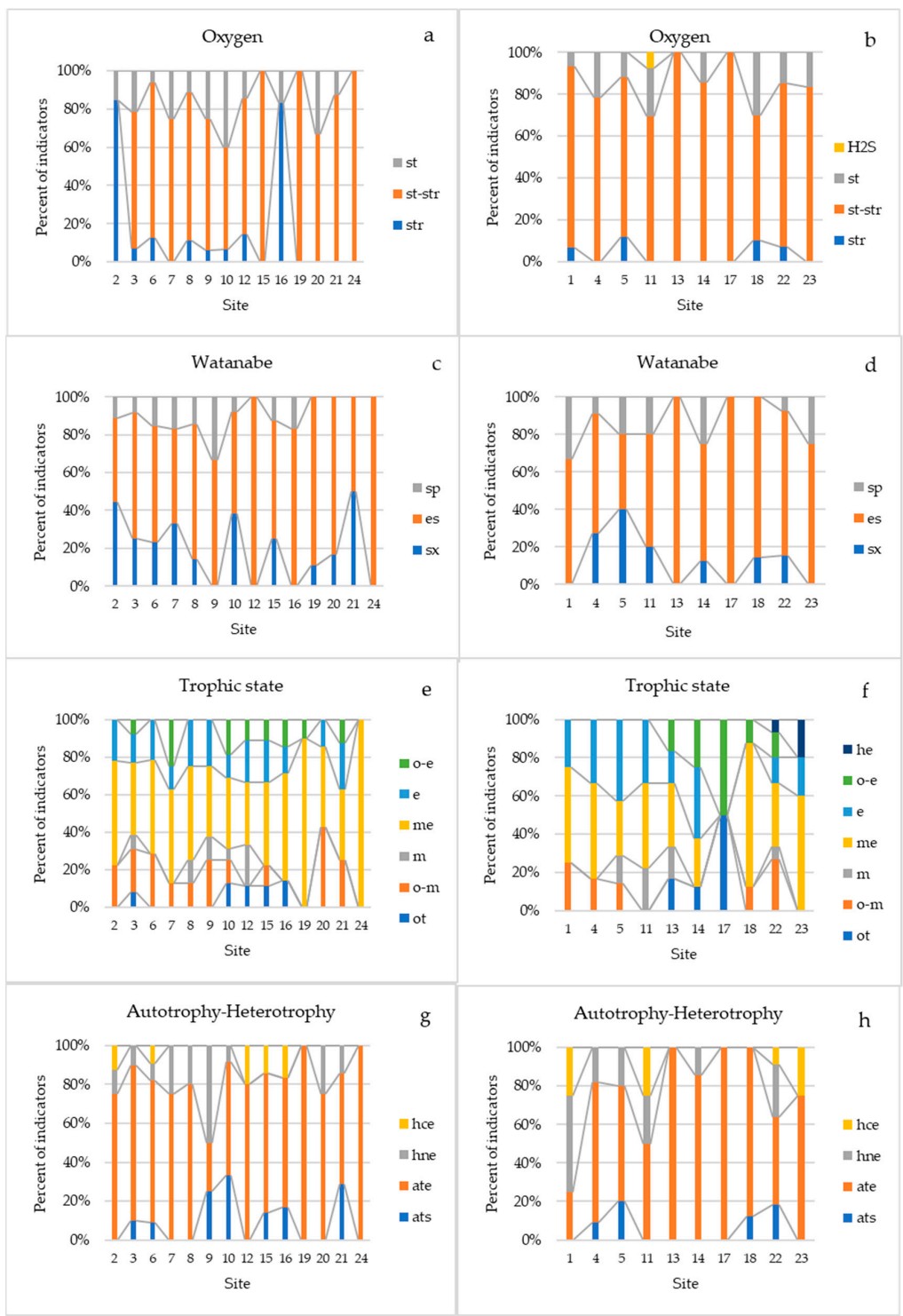

**Figure 8.** Ecological characteristics of phytoplankton species indicators (oxygenation, organic pollution by Watanabe, trophic state and nitrogen uptake metabolism) grouped according to the size of the rivers, S and M (**a**,**c**,**e**,**g**) and L and XL (**b**,**d**,**f**,**h**), in the Siversky Donets river basin (Ukrainian part). Abbreviations of ecological groups: by oxygenation, st—standing water, str—streaming water, st-str—low streaming water, and $H_2S$—sulfide indicators; by saprobity, sx—saproxenes; es—eurysaprobes; sp—saprophiles; by trophic state, ot—oligotraphentes, o-m—oligo-mesotraphhentes; m—mesotraphentes; me—meso-eutraphentes; e—eutraphentes; o-e—from oligo to eutraphentes; by nutrition type, ats—strict autotrophes; ate—autotrophic withstand low nitrogen load; hne—partial heterotrophes (mixotrophes); hce—permanent mixotrophes needing nitrogen supply.

The rivers grouped by their size with a focus on habitat preferences of algae do not show significant differences. A wider list of temperature, pH and salinity indicators is presented in the rivers of S and M size in contrast to L and XL rivers. Warm indicator species were present only in S and M size rivers (Figure 7c), whereas L and XL rivers did not have them at all (Figure 7d); additionally, bigger rivers contained more eterm species (with wider tolerance). These factors indicate that bigger rivers have colder temperatures, even having less cool-indicator species. The percentage of slightly alkaline water indicators (alf) is higher in the rivers of size S and M (Figure 7e), although these rivers also have acidophilous taxa surviving at pH 5–6 (acf). The salinity in the rivers of sites S and M varied more than in bigger rivers (Figure 7g), while the smaller rivers have halophobe (hb) species and the percentage of halophilous (hl) is higher.

The ecological status of the smaller (S, M) (Figure 8a,c,e,g) and bigger size rivers (L, XL) (Figure 8b,d,f,h) is presented by indicators of the following categories: oxygen amount, organic pollution (Watanabe), trophic state and nitrogen uptake metabolism. The ratio of running water indicators is higher in smaller rivers and, moreover, some representatives of hydrogen sulfide ($H_2S$) appear in the rivers of bigger size. The organic pollution (by Watanabe) reveals that smaller rivers have higher portion of saproxenous species (sx) and smaller amount of saprophilous species compared to the bigger rivers. It is revealed that the hypereutraphentic taxa are present in big rivers; there is also a higher portion of eutraphentic species compared to rivers of smaller size. As for the Nitrogen uptake metabolism indicators, the number of Nitrogen-autotrophic taxa tolerating very small concentrations of organically bound nitrogen (ats) is higher in smaller rivers.

Comparison of S and M rivers with L and XL rivers revealed the presence of I class indicator species, and a smaller number of indicators of III class indicators in the rivers of smaller size (Figure 9).

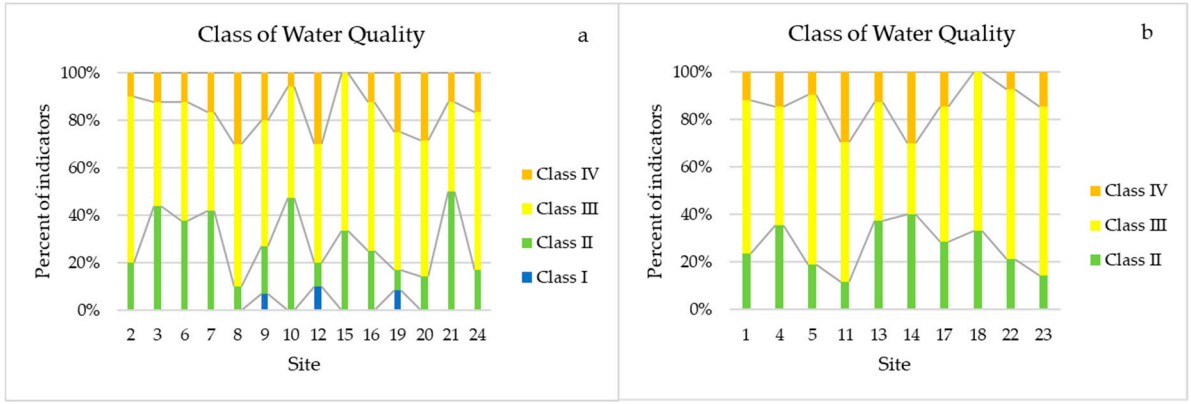

**Figure 9.** Water quality classes for S and M (**a**) and L and XL (**b**) rivers in the Siversky Donets river basin (Ukrainian part). Colors of water quality classes are presented according to the Water Framework Directive [14].

### 3.4. CCA Ordination Results for the Sampling Sites in the Siversky Donets River Basin

ANOVA-like Monte-Carlo permutation tests in CCA showed that the phytoplankton community structure was significantly related to several environmental variables (Tables 6 and 7, Figure 10). Specifically, the number of Chlorophyta species was related to higher $COD_{Mn}$ and $NO_2^-$ and lower $HCO_3^-$ and TDS. The number of Bacillariophyta species was related to higher COD and TDS, and lower $COD_{Mn}$ and $NO_2^-$. In contrast to Bacillariophyta, Cyanobacteria were more diverse at higher $COD_{Mn}$ and $HCO_3$-, and lower COD and TDS. Similarly, the total number of divisions was related to higher $COD_{Mn}$ and lower TDS and COD. The total number of species and families was related to lower $HCO_3^-$ and higher $NO_2^-$ and COD, but was independent from $COD_{Mn}$ and TDS. The saprobic index was strongly related to $HCO_3^-$.

**Table 6.** Ranking of environmental variables that significantly influenced the phytoplankton community structure based on the Monte Carlo permutation test in CCA.

| Environmental Variable | Variability Explained | *F*-Value | *p*-Value |
|---|---|---|---|
| $COD_{Mn}$ | 8.6 | 2.70 | 0.04 |
| TDS | 8.1 | 2.54 | 0.05 |
| $NO_2^-$ | 9.8 | 3.09 | 0.02 |
| $HCO_3^-$ | 10.3 | 3.25 | 0.02 |
| COD | 9.2 | 2.90 | 0.03 |

**Table 7.** Significance of the first two axes of the CCA model based on the Monte Carlo permutation test in CCA.

| | Variability Explained | *F*-Value | *p*-Value |
|---|---|---|---|
| Axis 1 | 24.4 | 7.67 | 0.007 |
| Axis 2 | 15.8 | 4.97 | 0.015 |

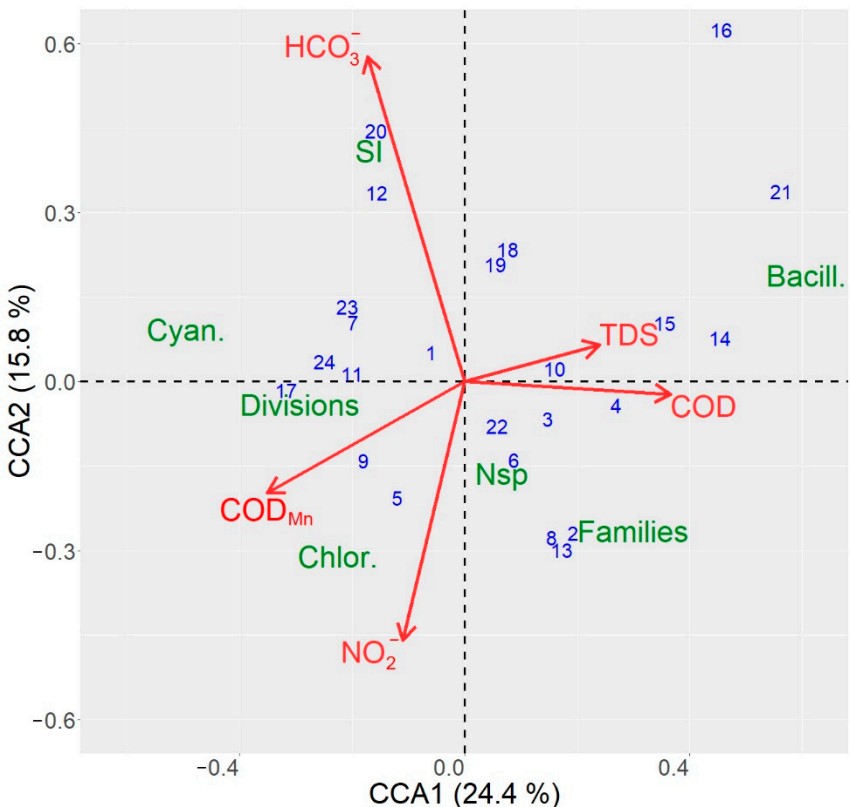

**Figure 10.** CCA ordination diagram of site scores, phytoplankton community structure parameters and selected environmental variables (represented by arrows). Only the significant explanatory environmental variables ($p < 0.05$) are presented.

## 4. Discussion

### 4.1. Chemical Contamination in the Sampling Sites

In general, increased mineralization was observed in most watercourses of the Siversky Donets basin. The highest values were observed in the Bychok River. It should be noted that the research was conducted in the baseflow period; thus, higher mineralization may have resulted from water supply from groundwater. Regarding the Bychok River, a right-bank tributary, it flows within the structural-denudation region of the Donetsk ridge, which

is composed of halogen and sulfate deposits, in the area of thick layers of rock salt (saline rocks); hence, the mineralization is also very high [12].

An important parameter of water quality is oxygenation, by which the production-destruction processes are assessed [64]. The results indicate uneven spatial distribution of the concentration of dissolved oxygen in the watercourses of the Seversky Donets basin, consistent with published data [65]. The excess oxygen saturation of water (130–155%) observed in the Bychok and Derkul rivers may be due to intensive photosynthesis. Conversely, the lower percentage of oxygen saturation, 30–43%, in the Mzha and Milova rivers was probably caused by the lower intensity of photosynthesis and higher intensity of chemical and biochemical oxidation of chemical compounds.

The highest values of permanganate oxidation of water and chemical oxygen demand, indicative of the presence of easily oxidizable organic and inorganic substances, were observed in the Bychok, Milova, Cherepakha, Derkul and Uda rivers. The increased water oxidizability indicates, on the one hand, an increase in the mineralization of organic matter of indigenous origin, and on the other hand, the inflow of organic and inorganic compounds with surface runoff, precipitation or insufficiently treated wastewater [66].

We found that the concentrations of both nitrogen and phosphorus compounds in the studied basin greatly fluctuated, which may be associated with different anthropogenic impacts on the catchment. The presence of dissolved ammonium in water may be a result of contamination by untreated or insufficiently treated wastewater. In addition, ammonium ions can be produced in the reservoir as a result of the organic matter mineralization. The high concentration of ammonium ions in the Udy River was most probably caused by the location of its catchment in the center of the highly economically developed and densely populated Kharkiv region. The river, especially its middle and lower sections, is known to have an increased level of pollution by industrial discharges, and, therefore, its waters are classified as polluted [67]. The high ammonium concentration in the section of the Siversky Donets river within the Kharkiv region can also be explained by its inflow with the waters of the river.

Nitrite ions are an intermediate stage in the nitrification/denitrification processes. Therefore, the concentrations of nitrite and nitrate ions are largely related to these processes and depend on the oxygen regime.

Low nitrite concentrations are usually characteristic of reservoirs and watercourses with a favorable oxygen regime. The higher values in the Siversky Donets river (Cheremushne) may be due to the presence of significant areas of washed away or eroded soils. Increased soil erosion in the river catchment is known to reduce the length of river network, siltation of springs, intensification of the chemical elements' migration from the soil and significant accumulation of bottom sediments [67]. To draw conclusions about the naturalness of the processes, it is necessary to analyze the proportion of nitrate ions in total amount of inorganic nitrogen compounds. Our calculations showed that the proportion of ammonium ions in water of the rivers Oskil, Derkul, Krasna (downstream of the village of Nyzhnya Duvanka) and Zherebets amounted to 23–35% of total inorganic nitrogen compounds. Nitrification processes lead to a 62 to 74% increase in the nitrate ions proportion in waters of these rivers, indicating persistent pollution over previous years.

The concentration of the intermediate link, nitrite ions, was uniformly low in all the above-mentioned rivers, approximately 3%. According to the literature data, nitrate ions dominate the rivers of the Donetsk ridge and their proportion varies from 71 to 80% [68]. However, in our study, the proportion of ammonium ions was high in Krasna (upstream of the village of Nyzhnya Duvanka) and Bychok rivers, 84 and 87%, respectively, and the proportion of nitrate ions was only 15% and 13%, respectively, probably indicating recent contamination or disturbance of the oxygen regime. In comparison to the literature data for the period 1993–2008, an almost two-fold increase in the proportion of ammonium ions was observed, from 55 to 94%, which may be related to the disturbance in nitrification processes, because the proportion of nitrate ions in that period have reached 40% [55] but in our study this was only 1%. The proportions of ammonium and nitrate ions in the water of

the Tetlyha and Siversky Donets rivers were almost equal, about 50%, which may indicate constant contamination of these waters with nitrogen-containing compounds.

Phosphorus compounds in surface waters are known to be more conservative than nitrogen [68]. Most of them are used by aquatic organisms and returned to the aquatic environment. The total concentration of dissolved phosphorus compounds in unpolluted natural waters varies from 0.005 to 0.2 mg P·L$^{-1}$, while in polluted waters, it can reach several milligrams per liter. The high concentration of phosphorus ions in the Udy river is most likely caused by high economic activity, as the river accepts wastewaters from Kharkiv city.

In summary, we can emphasize that the spatial dynamics of changes in hydrochemical parameters of water quality in the Seversky Donets basin depend primarily on geographical conditions and underlying rocks and only in some cases on anthropogenic load, which is consistent with the research of other authors [12,65,68]. With some precautions, this allows considering selected surface water massifs as references, except for the Uda River (the section of the Siverskyi Donets River within the Kharkiv region).

### 4.2. Characteristics of the Phytoplankton as a Biological Quality Element (BQE)

The basis of ecological monitoring is the Biological Quality Elements (BQE), and according to the Water Framework Directive [14], they are phytoplankton, phytobenthos, invertebrates, macrophytes and fishes [69]. For fulfillment of this work, a phytoplankton data analysis was made. In order to prove the reliability of ecological assessment based on the phytoplankton ecological characteristics for the Siversky Donets river basin, the results of state monitoring were used.

For the monitoring purposes phytoplankton sampling should be carried out once a year over the specified time interval, and comparative assessment should be done on the basis of results of single sampling [2]. We have analyzed data of the previous studies of the Siversky Donets river basin [25,27,67,70]. The results are close to those obtained in our survey, and thus, the latter were certainly used for the ecological status assessment.

Figure 11 was plotted on the basis of a comparative analysis of ecological state assessments based on phytoplankton, phytobenthos, macrophytes, invertebrates, and all BQEs in total, except for fishes, carried out on the basis of data of state monitoring of waters in the basin of Siversky Donets.

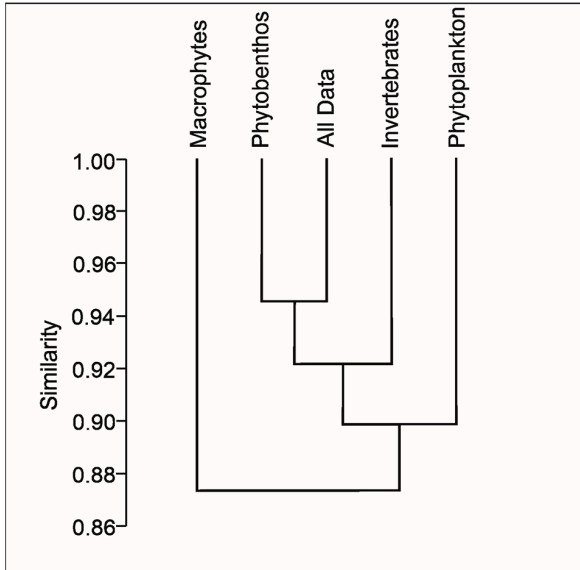

**Figure 11.** The similarity of the ecological state assessment results of the Siversky Donets river basin based on different biological quality elements (phytoplankton, macrophytes, phytobenthos, bottom invertebrates and integral assessment on the basis of all datasets).

The assessment based on phytobenthos was closest to the integrated assessment based on all BQEs, and the least accurate was the assessment on the basis of macrophytes. Verification of the assessments based on phytoplankton data revealed similarity to the assessment on the basis of the benthic invertebrates, phytobenthos and the whole BQE dataset. Figure 11 confirms the validity of the assessment according to phytoplankton data; the percentage of similarity of the phytoplankton assessments relative to the general data was at the level of 90%.

The phytoplankton in the Sirevsky Donets river basin is formed mostly by diatoms. This division prevalence is typical for the riverine phytoplankton [9,71–74] and is consistent with the literature data for the considered basin, in particular, the Siversky Donets river itself, which was characterized by predominance of Bacillariophyta (40%) and Chlorophyta (33%) [70]. It should be noted that many tychoplankton species may have been present in the water of studied rivers, which might be caused by high water flow combined with small depth of most and frequent drying out of some of the rivers, the factors not precisely documented in this study. Notwithstanding the high abundance (cell numbers) of Cyanobacteria, the biomass did not reach the values common for water "bloom" threshold [75]. As "bloom" events are one of the monitoring parameters, their absence is an important ecological fact [76] that should be considered in future investigations of the river basin. The presented list of the species with high abundance could be used by the water management authorities.

The indication of ecological state by some ecological characteristics of phytoplankton taxa showed interesting results that also may be used in the future assessment of the river basin. The temperature indicators revealed wider fluctuations in the rivers of S and M sizes compared to L and XL, which is logical as more water volume possess higher thermal capacity. Smaller rivers were more alkaline in contrast to the L and XL rivers, with one exception of the Udy river, for which acidification indicators are also characteristic of swampy waters. The salinity in the rivers of sites S and M have a wider list of indicators of salinity with the species that can be found in strictly fresh waters (hb) and species that sometimes can be found in saline waters (hl). This can be explained by the periodical desertification, the drying out of river basins.

The ecological state of the smaller rivers (S and M) is better than of bigger rivers (L and XL). The oxygenation and water mass dynamics were higher in smaller rivers then in bigger rivers, whereas the organic pollution (by Watanabe) was lower. The trophic state of the bigger rivers was higher than of smaller. The indicators of Nitrogen uptake metabolism of some groups revealed a transition from photosynthetic nutrition to heterotrophic nutrition, indicating a toxic effect in the bigger rivers.

Finally, the classes of water quality according to organic pollution by Sládeček showed that S and M rivers were cleaner than L and XL rivers.

### 4.3. Analysis of Species-Environment Relationships and Identification of Sites with the Highest Water Quality

In the present study, the three major divisions, Bacillariophyta, Chlorophyta, and Cyanobacteria, showed clearly different environmental requirements to achieve maximum diversity. Thus, high values of $COD_{Mn}$ and concentrations of $NO_2^-$ favored diversity of Chlorophyta. Diversity of Cyanobacteria was related to $HCO_3^-$ and $COD_{Mn}$. Other water pollution constituents, TDS and COD, related respectively to inorganic and organic pollution, favored high diversity of diatoms. $HCO_3^-$ is associated with alkalinity and buffering capacity of water needed for survival of wide spectrum of biological species, but its high amount along with $COD_{Mn}$ created preferable conditions for Cyanobacteria, probably serving as sources of excess inorganic and organic C. Moreover, increased alkalinity favors Cyanobacterial growth, repressing C uptake by other phytoplankton species [77]. A strong relationship was shown between quantitative parameters of Bacillariophyceae development and the values of total water mineralization and chemical oxygen demand [78–80]. Some diatoms are resistant to high water mineralization, e.g., *Cyclotella meneghiniana*, able to actively vegetate at 1100–3200 mg·$L^{-1}$ [80], consistent with the results of this study.

It can be supposed that low water quality is associated with conditions leading to the predominant growth of one or two of these groups. Therefore, we suggest that extreme values of essential environmental variables in this study led to imbalances in phytoplankton community structure indicating poor water quality. Based on this assumption, the sites closest to the origin point at the ordination diagram (see Figure 10) can be proposed as references with the highest water quality characterized by balanced phytoplankton composition and optimum values of the environmental variables. The distance of the sites from the CCA plot origin is presented in Table 8.

**Table 8.** Correlation of the major phytoplankton divisions diversity with distance from the CCA plot origin.

| Distance from CCA Plot Origin | Bacillariophyta | Chlorophyta | Cyanobacteria | Site (See Table 2) |
|---|---|---|---|---|
| 0.47 | 12 | 10 | 2 | 1 |
| 0.57 | 16 | 16 | 2 | 22 |
| 1.05 | 16 | 1 | 1 | 10 |
| 1.09 | 15 | 2 | 1 | 3 |
| 1.16 | 15 | 4 | 1 | 6 |
| 1.33 | 5 | 10 | 4 | 11 |
| 1.47 | 12 | 1 | 1 | 19 |
| 1.55 | 7 | 1 | 2 | 7 |
| 1.57 | 5 | 8 | 4 | 9 |
| 1.59 | 9 | 1 | 0 | 18 |
| 1.61 | 7 | 1 | 2 | 23 |
| 1.64 | 7 | 14 | 2 | 5 |
| 1.70 | 1 | 8 | 1 | 24 |
| 1.93 | 13 | 1 | 0 | 4 |
| 2.15 | 1 | 10 | 1 | 17 |
| 2.29 | 10 | 5 | 0 | 8 |
| 2.32 | 8 | 2 | 0 | 13 |
| 2.37 | 13 | 4 | 0 | 2 |
| 2.53 | 11 | 0 | 0 | 15 |
| 2.59 | 9 | 1 | 2 | 12 |
| 3.25 | 10 | 0 | 0 | 14 |
| 3.29 | 8 | 0 | 1 | 20 |
| 4.68 | 10 | 0 | 0 | 21 |
| 5.56 | 9 | 0 | 0 | 16 |
| Correlation with distance: | −0.19 | −0.50 ** | −0.48 * | |

*—statistically significant ($p < 0.05$); **—statistically significant ($p < 0.02$).

It is worth noting that the species number of the major phytoplankton divisions Bacillariophyta, Chlorophyta and Cyanobacteria [9,10] at individual sites inversely correlated with their distance from the CCA plot origin (see Table 8). The correlation was stronger in less presented divisions. Thus, Bacillariophyta species were in abundance and their diversity weakly correlated with the distance, while Chlorophyta and Cyanobacteria showed significant correlation (r = −0.50 and −0.48, respectively). We suppose that there are specific limiting conditions for the development of some Chlorophyta and Cyanobacteria species; therefore, their diversity can be a sensitive indicator of water quality in the considered region.

## 5. Conclusions

The analysis of phytoplankton data on the Siversky Donets river basin revealed diverse species composition with the predominance of diatoms. The biomass values of phytoplankton did not correspond to water "bloom", which should be considered in the future monitoring of the river basin.

The statistical data analyses revealed factors that control species diversity in the sites within Siversky Donets river basin with extreme values of essential environmental variables. The correlation between phytoplankton community structure and changes in water quality has been established.

Current study contributed to attaining the objectives of the study and identification of the reference sites with the highest water quality characterized by balanced phytoplankton composition and optimal values of the environmental variables. However, it should be noted that despite our efforts to select survey sites with minimal anthropogenic impact, some water bodies did not meet this requirement. The gathered information will be useful for the implementation of integrated approaches by water management authorities according to the basin principle in the Lower Don basin, to which the Siversky Donets basin belongs. However, since the Siversky Donets basin is one of the least studied among the main river basins of Ukraine, further studies of its hydrobiological status are needed.

**Supplementary Materials:** The following are available online at https://www.mdpi.com/article/10.3390/w13233368/s1, Table S1: Species composition of the phytoplankton in the Siversky Donets River Basin.

**Author Contributions:** Conceptualization of this study was done by O.B. and S.A.; the investigation was made by O.B., O.L., O.M. and M.P.; methodology was prepared by O.L., O.B., O.M. and S.A.; statistics and mapping were prepared by O.P.; the manuscript was written and edited by O.B., S.A., S.B., O.M., I.N., M.P. and O.P. All authors have read and agreed to the published version of the manuscript.

**Funding:** This research was partially funded by the grant of the National Academy of Sciences of Ukraine to research laboratories/groups of young scientists to conduct research in priority areas of science and technology (code 6541230, Project State Reg. No. 0118U005432), Target complex interdisciplinary research program of the National Academy of Sciences of Ukraine on sustainable development and environmental management under global environmental changes for 2020–2024 (Project State Reg. No. 0120U103140), and OSCE Project "Helping Expand an Environmental Monitoring System in Donbas" (No. UB 3200444).

**Data Availability Statement:** The data presented in this study are available on request from the corresponding author.

**Acknowledgments:** The authors acknowledge financial and organizational support given by the Institute of Hydrobiology of the National Academy of Science of Ukraine and financial support from OSCE. We are also grateful to Igor Abramiuk and Oleg Golub (Institute of hydrobiology of NASU) for their assistance in the field surveys to the Eastern part of Ukraine and the Siversky Donets river basin, and Marya Linchuk (Institute of hydrobiology of NASU) for the provided hydrochemical data. Additionally, the authors are very grateful to Andrii Tarieiev for proofreading the text.

**Conflicts of Interest:** The authors declare no conflict of interest.

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
