# Peer review of "Preliminary Assessment of Ecological Status of the Siversky Donets River Basin (Ukraine) Based on Phytoplankton Parameters and Its Verification by Other Biological Data"

_water, doi:10.3390/w13233368_

Round 1

Reviewer 1 Report

Instaed of the present title of MS, I propose the following one: Preliminary assessment of ecological status of the Siversky Donets river basin (Ukraine) based on phytoplankton parameters and its verification by other biological data - (because authors were collected at a total of 24 sampling points and only once)

The aim (s) described in lines 81-85 of the introduction should formulated by the authors into separate points.

According to the Sampling strategy subchapter, authors were collected at a total of 24 sampling points and only once. This is not sufficient for a thorough analysis, even if many physicochemical parameters have been studied and a “complete” phytoplankton analysis has been performed.

It would be important to provide a summary table of species found in the research. This could list sampling sites and ecological preferences (see subchapter 3.3), etc. For example, in Figure 2, Bacillariophyceae makes up 63% of the species (there are usually only a few dozen euplanktonic diatoms in a river).

Most of the rivers studied here, have high TDS - their water is turbid or very turbid, where due to the unfavorable underwater light climate there are poor phytoplankton (euplanktonic species) and many tichoplankton (strong current drifts them off the periphyton). It should be known what the actual water flow (current) was at the time of sampling (collection was at low water, medium water or high water).

In the case of phytoplankton, in addition to quantity (cell-number and biomass), chlorophyll-a is also very important. Knowing this, the ecological status of the river can be determined more precisely.

There is little (not enough) comparison with international literature concerning on river phytoplankton in the discussion.

Proposed references:

Abonyi, A., É. Ács, A. Hidas, I. Grigorszky, G. Várbíró, G. Borics & K. T. Kiss, 2018. Functional diversity of phytoplankton highlights long-term gradual regime shift in the middle section of the Danube River due to global warming, human impacts and oligotrophication. Freshwater Biology, 63(5):456-472 doi:10.1111/fwb.13084.

Bolgovics, Á ; Várbíró, G ; Ács, É ; Trábert, Z ; Kiss, KT ; Pozderka, V ; Görgényi, J ; Boda, P ; Lukács, B-A ; Nagy-László, Z et al. (2017) Phytoplankton of rhithral rivers: its origin, diversity and possible use for quality-assessment. - Ecological Indicators, Volume 81, October 2017, Pages 587-596, https://doi.org/10.1016/j.ecolind.2017.04.052

Borics, G., G. Várbíró, I. Grigorszky, E. Krasznai, S. Szabó and K. T. Kiss. 2007. A new evaluation of potamo-plankton for the assessmant of the ecological status of rivers. - Arch Hydrobiol. Suppl. 161/3-4, Large Rivers 17: 465-486.

Martin T. Dokulil, 2014. Phytoplankton of the River Danube: Composition, Seasonality and Long-Term Dynamics. – In: I. Liska (ed.), The Danube River Basin, Hdb Env Chem, DOI 10.1007/698_2014_293, © Springer-Verlag Berlin Heidelberg 2014.

Martin T. Dokulil, Ulrich Donabaum, (2014). Phytoplankton of the Danube River: Composition and Long-Term Dynamics. - Acta zool. bulg., Suppl. 7, 2014: 147-152

Martin Dokulil, Ulrich Donabaum (2015) 8. Phytoplankton. – In: Joint Danube Survey 3, A Comprehensive Analysis of Danube Water QualityICPDR / International Commission for the Protection of the Danube River / www.icpdr.org, pp: 119-125.

Rusanov et al. (2021) Relative importance of climate and spatial processes in shaping species composition, functional structure and beta diversity of phytoplankton in a large river. - Science of The Total Environment 807:150891, DOI: 10.1016/j.scitotenv.2021.150891

Paiva Da Silva, José Etham De Lucena BarbosaLucineide Maria SantanaLucineide Maria SantanaLuciana BarbosaLuciana Barbosa. (2021) Phytoplankton functional groups in shallow aquatic ecosystems from the semiarid region of Brazil Grupos funcionais fitoplanctônicos em ecossistemas aquáticos rasos na região semiárida do Brasil. - Acta Limnologica Brasiliensia 33:e24, DOI: 10.1590/S2179-975X10320

Igor Stanković, Marija Gligora Udovič, Gábor Borics (2021) 08 Phytoplankton. – In: Liška et al. (eds.), Joint Danube Survey 4. Scientific Report: www.danubesurvey.org, A shared analysis of the Danube River, pp: 73-82. 

Author Response

Dear Reviewer1,

Thank you so much for deep analysis of the manuscript, your ideas and comments. These helped us to improve the paper. Some of the comments needs an explanation, that could be found below.

Instead of the present title of MS, I propose the following one: Preliminary assessment of ecological status of the Siversky Donets river basin (Ukraine) based on phytoplankton parameters and its verification by other biological data - (because authors were collected at a total of 24 sampling points and only once)

Response: The title is changed according to the proposed.

The aim (s) described in lines 81-85 of the introduction should formulated by the authors into separate points.

Response: The aim was reformulated

According to the Sampling strategy subchapter, authors were collected at a total of 24 sampling points and only once. This is not sufficient for a thorough analysis, even if many physicochemical parameters have been studied and a “complete” phytoplankton analysis has been performed.

Response: Dear reviewer, the Siversky basin is the first basin where the national monitoring is started, the information obtained during first survey is very valuable and should be published. The obtained data has never been analyzed and never been obtained in such full data selection and will be used by scientists and authorities in future. We have changed the title and hope this will show that the obtained data is only the start of fully monitoring

It would be important to provide a summary table of species found in the research. This could list sampling sites and ecological preferences (see subchapter 3.3), etc. For example, in Figure 2, Bacillariophyceae makes up 63% of the species (there are usually only a few dozen euplanktonic diatoms in a river).

Response: The additional table is ready and will be presented in the paper as separate file (Supplement). Moreover, the dominant species are also in the table in the text.

Most of the rivers studied here, have high TDS - their water is turbid or very turbid, where due to the unfavourable underwater light climate there are poor phytoplankton (euplanktonic species) and many tichoplankton (strong current drifts them off the periphyton). It should be known what the actual water flow (current) was at the time of sampling (collection was at low water, medium water or high water).

Response: We realize, that in the rivers were numbers of tychoplankton species, and the reason can be also different factors forcing for instance periphyton freely floating in the river. The current was not measured for this study, however also small depth most of the rivers as well as drying out could force the tychoplankton species’ appearing. During next studies these factors will be measured and analysed.

In the case of phytoplankton, in addition to quantity (cell-number and biomass), chlorophyll-a is also very important. Knowing this, the ecological status of the river can be determined more precisely.

Response: You are absolutely right, this parameter will be included in future monitoring, Now the biomass was calculated due to equating the cells to specific geometrical forms according to Hillebrand et al., 1999 and this also can be additional parameter testifying condition of phytoplankton instead of chlorophyll-a.

There is little (not enough) comparison with international literature concerning on river phytoplankton in the discussion.

Response: Thank you for the provided references, they were added to the discussion and into the other parts of the paper.

Proposed references:

Abonyi, A., É. Ács, A. Hidas, I. Grigorszky, G. Várbíró, G. Borics & K. T. Kiss, 2018. Functional diversity of phytoplankton highlights long-term gradual regime shift in the middle section of the Danube River due to global warming, human impacts and oligotrophication. Freshwater Biology, 63(5):456-472 doi:10.1111/fwb.13084.

Bolgovics, Á ; Várbíró, G ; Ács, É ; Trábert, Z ; Kiss, KT ; Pozderka, V ; Görgényi, J ; Boda, P ; Lukács, B-A ; Nagy-László, Z et al. (2017) Phytoplankton of rhithral rivers: its origin, diversity and possible use for quality-assessment. - Ecological Indicators, Volume 81, October 2017, Pages 587-596, https://doi.org/10.1016/j.ecolind.2017.04.052

Borics, G., G. Várbíró, I. Grigorszky, E. Krasznai, S. Szabó and K. T. Kiss. 2007. A new evaluation of potamo-plankton for the assessmant of the ecological status of rivers. - Arch Hydrobiol. Suppl. 161/3-4, Large Rivers 17: 465-486.

Martin T. Dokulil, 2014. Phytoplankton of the River Danube: Composition, Seasonality and Long-Term Dynamics. – In: I. Liska (ed.), The Danube River Basin, Hdb Env Chem, DOI 10.1007/698_2014_293, © Springer-Verlag Berlin Heidelberg 2014.

Martin T. Dokulil, Ulrich Donabaum, (2014). Phytoplankton of the Danube River: Composition and Long-Term Dynamics. - Acta zool. bulg., Suppl. 7, 2014: 147-152

Martin Dokulil, Ulrich Donabaum (2015) 8. Phytoplankton. – In: Joint Danube Survey 3, A Comprehensive Analysis of Danube Water QualityICPDR / International Commission for the Protection of the Danube River / www.icpdr.org, pp: 119-125.

Rusanov et al. (2021) Relative importance of climate and spatial processes in shaping species composition, functional structure and beta diversity of phytoplankton in a large river. - Science of The Total Environment 807:150891, DOI: 10.1016/j.scitotenv.2021.150891

Paiva Da Silva, José Etham De Lucena BarbosaLucineide Maria SantanaLucineide Maria SantanaLuciana BarbosaLuciana Barbosa. (2021) Phytoplankton functional groups in shallow aquatic ecosystems from the semiarid region of Brazil Grupos funcionais fitoplanctônicos em ecossistemas aquáticos rasos na região semiárida do Brasil. - Acta Limnologica Brasiliensia 33:e24, DOI: 10.1590/S2179-975X10320

Igor Stanković, Marija Gligora Udovič, Gábor Borics (2021) 08 Phytoplankton. – In: Liška et al. (eds.), Joint Danube Survey 4. Scientific Report: www.danubesurvey.org, A shared analysis of the Danube River, pp: 73-82. 

As for the points highlighted in the pdf

this can be minus 7 centigrades

Response: It is true temperature, checked in the reference. It is a medium month temperature and it is really so.

This paragraph is not clear to the reviewer

Response: The 2 paragraphs were included by a mistake. Deleted.

From the manuscript: “this is an objective with 10x magnification which is not enough for determination of phytoplankton species smaller than ~ 15-20 micrometer, used a Nageotte chamber”

Response: The 10 referred to ocular lens, now it is changed to 40x as objective magnification so as not to confuse the reader.

the most easily uptaken nitrogen source for phytoplankton!!!

Response: The data are presented as minimum and maximum values, we do not say that this amount of nitrogen is extra high, we just state that it was maximum.

many of the data described in the following paragraphs (235-286 lines) would be better presented in figures and write only a short summary about the important components

Response: Dear reviewer, we just highlighted the minimum and maximum values, all data are presented in the table. The important components are described in the statistical analysis section. In turn, we have rewritten some parts in Chemical section.

It would be important to provide an additional table of the species (woth species name) found in this research. For example, in Figure 2, Bacillariophyceae makes up 63% of the species (there are usually only a few dozen euplanktonic diatoms in a river). The Table 3 is not informative in this respect.

Response: Additional information now is presented in the Supplement.

Instead of % composition, absolute species number data would be much more informative to compare the two groups (S,M and L, LX).

Response: We tried this idea, before presenting the results the way they are. However, the graphs were not informative, as it is only preliminary investigations. In this case, the percentage revealed to be more informative.

It would be better to include the data in the paragraph in a table instead of the current textual list, with species names (indicating which is euplanktonic and tichoplanktonic) and division names, as well as numbers of sampling sites. Thus, Figure 5 would be more understandable.

Response: Thank you for this idea, the data are organed in the way you recommend and the Tables are provided.

there are two scales on left and right site of graph, but no legend

Response: Only Cyanobacteria is presented by additional scale, because their values differ from other groups significantly and it was the only possible way to make this group visible on the graph.

It would be better to include the data in the paragraph in a table instead of the current textual list, with species names (indicating which is euplanktonic and tichoplanktonic) and division names, as well as numbers of sampling sites. Thus, Figure 6 would be more understandable.

Response: Done, thank you for the idea.

these abbreviations are not commonly known, so explain them please.

Response: Necessary part inserted to the text (Materials and Methods section)

what does it means: oxygen waters?

Response: Changed in the text into “species indicating waters with moderate dynamics and oxygenation”

It would be better to include the data in the paragraph in a table instead of the current textual list,

Response: For illustrating these data, the Figures 7 and 8 are presented. The text only helps the reader to understand what is shown on pictures. In turn, the text has more than 7 tables and also supplement, we consider that the text for presenting this data is better option.

Temperature of different rivers varied between 14,5 - 26 centigrades. On Fig.b. (temp) only the river 4 has cold preferent algae - in text is written something else.

“The smaller rivers have more warm-indicator taxa; whereas number of cold indicators is higher in bigger rivers.”

Response: The statement is true. Small rivers have warm-indicator species, whereas L and XL rivers do not have them at all, also bigger rivers have more eterm species (with wider tolerance). All these factors reveal that bigger rivers have colder temperatures, even having less cool-indicators species. And yes, saying cool-indicators we use data from the Barinova et al., 2019 source where the indicators shown with the rage you name.

abbreviations of different preferences are not commonly known, it should write them on figure legend

Response: we have added this information into the Materials and Methods section

This figure and the paragraphs before and after it seem even more relevant to the chapter results.

Response: The data that were used here are from literature data and was used by us as a Discussion to underline that phytoplankton is the good indicator of water quality as other biological parameters.

Dear reviewer, also other your ideas with citing literature was considered and appropriate changes were made.

Reviewer 2 Report

The whole manuscript should be thoroughly revised. In the abstract, authors insert contractions which are not explained and abstract should be written without contractions. 

Parts of the paper which deal with nutrients determination should be thoroughly revised and corrected. Some of the statements are not chemically correct. The term "concentration" should be used instead of "content" and  the right term for specific nutrients should be used. For example, "Phosphorus" doesn't mean anything. Is it organic phosphorous? Or particulate? It should be written clearly, straightforwardly and uniformly.

Please revise, and insert two sentences about nutrient determination. If the method was standard, then at least a method name or a method number should be given.

Authors introduce contractions without explanation, explanations are given only later in the text. 

Author often use active form, possessive and personal pronouns which should be avoided in scientific writing: 201-204; 280...

161: UK pump? Is it a pump from the Brittany, or does UK stands for the producer. Or for the type? Needs clarification. 

113-120: What is this? Does this belong to the manuscript?

164-166: What are those "common" methods? There is no reference? Needs clarification. 

280-286: not clear, revise.

295-297: not clear. What does WB stands for? Water body, perhaps? Needs clarification.

319: Grammar!! "This species was..." is incorrect! "These species were" or "this specie is. "

325. Should be revised! 

358-399: Needs clarification. Very hard to read and follow with all these personal names in brackets and outside brackets.

412-417: Revise!

417: English style is bad. Species afford? Should be revised.

441: Grammar!

504: "Presence of ammonium compounds in water?" It is incorrect. Dissolved ammonium is determined not ammonium compounds.   

Please, revise the whole manuscript with thorough English proofing before resubmitting. In the current form, it is very hard to read and almost impossible to understand. Chemical part is very badly written.   

Author Response

Dear Reviewer 2,

Thank you so much for deep analysis of the manuscript, your ideas and comments. These helped us to improve the paper. Some of the comments needs an explanation that could be found below.

The whole manuscript should be thoroughly revised. In the abstract, authors insert contractions which are not explained and abstract should be written without contractions. 

Response: Thank you so much for this comment. We gave our manuscript to native speaker and now manuscript significantly improved.

Parts of the paper which deal with nutrients determination should be thoroughly revised and corrected. Some of the statements are not chemically correct. The term "concentration" should be used instead of "content" and  the right term for specific nutrients should be used. For example, "Phosphorus" doesn't mean anything. Is it organic phosphorous? Or particulate? It should be written clearly, straightforwardly and uniformly.

Response: Changed and corrected. Thank you so much for the ideas.

Please revise, and insert two sentences about nutrient determination. If the method was standard, then at least a method name or a method number should be given.

Authors introduce contractions without explanation, explanations are given only later in the text.

Response: Changed 

Author often use active form, possessive and personal pronouns which should be avoided in scientific writing: 201-204; 280...

Response: There are no strict rules of using active or passive voice in scientific wrtiting but only recommendation that both should be balanced (https://www.nature.com/articles/d41586-018-02404-4?fbclid=IwAR35D3A0P0xozQvFHUWWxPIaB8g91_A7xIIQ9hmBFxQMFdVArp4P8ztmSYk and https://sites.duke.edu/scientificwriting/passive-voice-in-scientific-writing/?fbclid=IwAR3JudYT3e_e8CDsof5Kywtp2_3PlfoNnr6eOG7iNLjJBtR5DQPcLtAXOUE https://advice.writing.utoronto.ca/types-of-writing/active-voice-in-science/?fbclid=IwAR26YcQHHLqGTSQhbvW0TIM8GPZCvkbwgU0IfE1ruMTOpqeT4BHIMTblXLs ). We used active voice with the aim to make our manuscript more clear, concise and easy to read for the readers. However, we also used passive voice extensively.

161: UK pump? Is it a pump from the Brittany, or does UK stands for the producer. Or for the type? Needs clarification. 

Response: Changed into clear meaning: сompressor unit 40-2M (M-Apparatura, Ukraine).

113-120: What is this? Does this belong to the manuscript?

Response: Deleted

164-166: What are those "common" methods? There is no reference? Needs clarification. 

Response: The appropriate source was added into the text.

Osadchy, V.I.; Nabyvanets, B.Y.; Osadcha, N.M.; Nabyvanets, Y.B. Hydrochemical reference book. Surface waters of Ukraine. Hydrochemical calculations. Methods of analysis. Nika-Center, Kyiv, 2008; pp. 1655. (In Ukrainian).

280-286: not clear, revise.

Response: This part was revised

295-297: not clear. What does WB stands for? Water body, perhaps? Needs clarification.

Response: explained

319: Grammar!! "This species was..." is incorrect! "These species were" or "this specie is. "

Response: Changed into taxon to show this is singular

  1. Should be revised! 

Response: revised

358-399: Needs clarification. Very hard to read and follow with all these personal names in brackets and outside brackets.

Response: The additional table was added into the text

412-417: Revise!

Response: revised

417: English style is bad. Species afford? Should be revised.

Response: revised

441: Grammar!

Response: revised

504: "Presence of ammonium compounds in water?" It is incorrect. Dissolved ammonium is determined not ammonium compounds.   

Response: Changed into dissolved ammonium

Please, revise the whole manuscript with thorough English proofing before resubmitting. In the current form, it is very hard to read and almost impossible to understand. Chemical part is very badly written.   

Response: Chemical part is rewritten, also the text was proofread by the English native speaker.

As for the points highlighted in the pdf

discussing about "blooming parameters" should be based on biomass and not cell number of phytoplankton.

Response: You are absolutely right, changed it according to biomass values.

Dear reviewer, also other your ideas with citing literature was considering and appropriate changes were made.

Round 2

Reviewer 1 Report

Manuscript corrections additions are acceptable.

Author Response

Dear Reviewer1,

Thank you so much for deep analysis of the manuscript, your ideas and comments. These helped us to improve the paper. Some of the comments needs an explanation, that could be found below.

Instead of the present title of MS, I propose the following one: Preliminary assessment of ecological status of the Siversky Donets river basin (Ukraine) based on phytoplankton parameters and its verification by other biological data - (because authors were collected at a total of 24 sampling points and only once)

Response: The title is changed according to the proposed.

The aim (s) described in lines 81-85 of the introduction should formulated by the authors into separate points.

Response: The aim was reformulated

According to the Sampling strategy subchapter, authors were collected at a total of 24 sampling points and only once. This is not sufficient for a thorough analysis, even if many physicochemical parameters have been studied and a “complete” phytoplankton analysis has been performed.

Response: Dear reviewer, the Siversky basin is the first basin where the national monitoring is started, the information obtained during first survey is very valuable and should be published. The obtained data has never been analyzed and never been obtained in such full data selection and will be used by scientists and authorities in future. We have changed the title and hope this will show that the obtained data is only the start of complete monitoring

It would be important to provide a summary table of species found in the research. This could list sampling sites and ecological preferences (see subchapter 3.3), etc. For example, in Figure 2, Bacillariophyceae makes up 63% of the species (there are usually only a few dozen euplanktonic diatoms in a river).

Response: The additional table is ready and will be presented in the paper as separate file (Supplement). Moreover, the dominant species are also in the table in the text.

Most of the rivers studied here, have high TDS - their water is turbid or very turbid, where due to the unfavourable underwater light climate there are poor phytoplankton (euplanktonic species) and many tichoplankton (strong current drifts them off the periphyton). It should be known what the actual water flow (current) was at the time of sampling (collection was at low water, medium water or high water).

Response: We realize that many tychoplankton species are present in the studied rivers, because of the small depth of most of them and frequent drying out of some of them. The current was not measured in this study, as well as the other factors, but they were at approximately the same level for all the rivers. We have added a snippet acknowledging this fact in subsection 4.2: “It should be noted that many tychoplankton species may have been present in the water of studied rivers, which might be caused by high water flow combined with small depth of most and frequent drying out of some of the rivers, the factors not precisely documented in this study.”

In the case of phytoplankton, in addition to quantity (cell-number and biomass), chlorophyll-a is also very important. Knowing this, the ecological status of the river can be determined more precisely.

Response: You are absolutely right, this parameter will be included in future monitoring, Now the biomass was calculated due to equating the cells to specific geometrical forms according to Hillebrand et al., 1999 and this also can be additional parameter indicating the state of phytoplankton instead of chlorophyll-a. Also, we have added an analysis of published data regarding chlorophyll a concentration in the studied rivers.

There is little (not enough) comparison with international literature concerning on river phytoplankton in the discussion.

Response: Thank you for the provided references, they were added to the discussion and into the other parts of the paper.

Proposed references:

Abonyi, A., É. Ács, A. Hidas, I. Grigorszky, G. Várbíró, G. Borics & K. T. Kiss, 2018. Functional diversity of phytoplankton highlights long-term gradual regime shift in the middle section of the Danube River due to global warming, human impacts and oligotrophication. Freshwater Biology, 63(5):456-472 doi:10.1111/fwb.13084.

Bolgovics, Á ; Várbíró, G ; Ács, É ; Trábert, Z ; Kiss, KT ; Pozderka, V ; Görgényi, J ; Boda, P ; Lukács, B-A ; Nagy-László, Z et al. (2017) Phytoplankton of rhithral rivers: its origin, diversity and possible use for quality-assessment. - Ecological Indicators, Volume 81, October 2017, Pages 587-596, https://doi.org/10.1016/j.ecolind.2017.04.052

Borics, G., G. Várbíró, I. Grigorszky, E. Krasznai, S. Szabó and K. T. Kiss. 2007. A new evaluation of potamo-plankton for the assessmant of the ecological status of rivers. - Arch Hydrobiol. Suppl. 161/3-4, Large Rivers 17: 465-486.

Martin T. Dokulil, 2014. Phytoplankton of the River Danube: Composition, Seasonality and Long-Term Dynamics. – In: I. Liska (ed.), The Danube River Basin, Hdb Env Chem, DOI 10.1007/698_2014_293, © Springer-Verlag Berlin Heidelberg 2014.

Martin T. Dokulil, Ulrich Donabaum, (2014). Phytoplankton of the Danube River: Composition and Long-Term Dynamics. - Acta zool. bulg., Suppl. 7, 2014: 147-152

Martin Dokulil, Ulrich Donabaum (2015) 8. Phytoplankton. – In: Joint Danube Survey 3, A Comprehensive Analysis of Danube Water QualityICPDR / International Commission for the Protection of the Danube River / www.icpdr.org, pp: 119-125.

Rusanov et al. (2021) Relative importance of climate and spatial processes in shaping species composition, functional structure and beta diversity of phytoplankton in a large river. - Science of The Total Environment 807:150891, DOI: 10.1016/j.scitotenv.2021.150891

Paiva Da Silva, José Etham De Lucena BarbosaLucineide Maria SantanaLucineide Maria SantanaLuciana BarbosaLuciana Barbosa. (2021) Phytoplankton functional groups in shallow aquatic ecosystems from the semiarid region of Brazil Grupos funcionais fitoplanctônicos em ecossistemas aquáticos rasos na região semiárida do Brasil. - Acta Limnologica Brasiliensia 33:e24, DOI: 10.1590/S2179-975X10320

Igor Stanković, Marija Gligora Udovič, Gábor Borics (2021) 08 Phytoplankton. – In: Liška et al. (eds.), Joint Danube Survey 4. Scientific Report: www.danubesurvey.org, A shared analysis of the Danube River, pp: 73-82. 

As for the points highlighted in the pdf

this can be minus 7 centigrades

Response: It is true temperature, checked in the reference. It is a medium month temperature and it is really so.

This paragraph is not clear to the reviewer

Response: The 2 paragraphs were included by a mistake. Deleted.

From the manuscript: “this is an objective with 10x magnification which is not enough for determination of phytoplankton species smaller than ~ 15-20 micrometer, used a Nageotte chamber”

Response: The 10 referred to ocular lens, now it is changed to 40x as objective magnification so as not to confuse the reader.

the most easily uptaken nitrogen source for phytoplankton!!!

Response: We are not sure we understood the comment, could you please explain it. The data are presented as minimum and maximum values, we do not say that this amount of nitrogen is extra high, we just state that it was maximum. Also, we have added a statement that dissolved ammonium is the most easily uptaken nitrogen source for phytoplankton.

many of the data described in the following paragraphs (235-286 lines) would be better presented in figures and write only a short summary about the important components

Response: Dear reviewer, we have rewrited the text in the “Results” section to show anly maximum and minimum values and highligh some peculiarities. The important components are described in the statistical analysis section. In turn, we have rewritten some parts in Chemical section.

It would be important to provide an additional table of the species (woth species name) found in this research. For example, in Figure 2, Bacillariophyceae makes up 63% of the species (there are usually only a few dozen euplanktonic diatoms in a river). The Table 3 is not informative in this respect.

Response: Additional information now is presented in the Supplement.

Instead of % composition, absolute species number data would be much more informative to compare the two groups (S,M and L, LX).

Response: We tried this idea, before presenting the results the way they are. However, the graphs were not informative, as it is only preliminary investigations. In this case, the percentage revealed to be more informative.

It would be better to include the data in the paragraph in a table instead of the current textual list, with species names (indicating which is euplanktonic and tichoplanktonic) and division names, as well as numbers of sampling sites. Thus, Figure 5 would be more understandable.

Response: Thank you for this idea, the data are organized in the way you recommend and the Tables are provided.

there are two scales on left and right site of graph, but no legend

Response: Only Cyanobacteria is presented by additional scale, because their values differ from other groups significantly and it was the only possible way to make this group visible on the graph. On the legend, we have written “Cyanobacteria (additional axis)”. Also, we have marked the axis “N(cyan), cells*103*L-1”.

It would be better to include the data in the paragraph in a table instead of the current textual list, with species names (indicating which is euplanktonic and tichoplanktonic) and division names, as well as numbers of sampling sites. Thus, Figure 6 would be more understandable.

Response: We have added these tables. However, we have not indicated which species is euplanktonic and tichoplanktonic because we have not found a precise classification.

these abbreviations are not commonly known, so explain them please.

Response: Necessary part inserted to the text (Materials and Methods section)

what does it means: oxygen waters?

Response: Changed in the text into “species indicating waters with moderate dynamics and oxygenation”

It would be better to include the data in the paragraph in a table instead of the current textual list,

Response: For illustrating these data, the Figures 7 and 8 are presented. The text helps the reader to understand what is shown in pictures. In turn, the text has more than 7 tables and also supplement, we consider that the text for presenting this data is better option.

Temperature of different rivers varied between 14,5 - 26 centigrades. On Fig.b. (temp) only the river 4 has cold preferent algae - in text is written something else.

“The smaller rivers have more warm-indicator taxa; whereas number of cold indicators is higher in bigger rivers.”

Response: We have slightly changed the text. Now it reads as follows: “Warm indicator species were present only in S and M size rivers, whereas L and XL rivers did not have them at all, also bigger rivers contained more eterm species (with wider tol-erance). These factors indicate that bigger rivers have colder temperatures, even having less cool-indicators species.” The rationale is that small rivers have warm-indicator species, whereas L and XL rivers do not have them at all, also bigger rivers have more eterm species (with wider tolerance). All these factors reveal that bigger rivers have colder temperatures, even having less cool-indicators species. Saying cool-indicators we use data from the Barinova et al., 2019 source where the indicators shown with the rage you name.

abbreviations of different preferences are not commonly known, it should write them on figure legend

Response: we have added this information into the Materials and Methods section and in the figure legend

This figure and the paragraphs before and after it seem even more relevant to the chapter results.

Response: The data that were used here are from literature data and was used by us as a Discussion to underline that phytoplankton is the good indicator of water quality as other biological parameters.

Dear reviewer, other your ideas with citing literature were also considered and appropriate changes were ma

Reviewer 2 Report

After the revision, manuscript is readable and understandable. Thank you for the revision of the chemical part. Quality of the manuscript is greatly improved and represents a worthy contribution to the specific scientific field.  

Author Response

Dear Reviewer 2,

Thank you so much for deep analysis of the manuscript, your comments are accepted in our ms. It helped us to improve the paper. 

With best regards,

The Authors